# Columnar neurons support saccadic bar tracking in *Drosophila*

**Giovanni Frighetto, Mark A Frye\***

Department of Integrative Biology and Physiology, University of California, Los Angeles, Los Angeles, United States

**Abstract** Tracking visual objects while maintaining stable gaze is complicated by the different computational requirements for figure-ground discrimination, and the distinct behaviors that these computations coordinate. *Drosophila melanogaster* uses smooth optomotor head and body movements to stabilize gaze, and impulsive saccades to pursue elongated vertical bars. Directionally selective motion detectors T4 and T5 cells provide inputs to large-field neurons in the lobula plate, which control optomotor gaze stabilization behavior. Here, we hypothesized that an anatomically parallel pathway represented by T3 cells, which provide inputs to the lobula, drives bar tracking body saccades. We combined physiological and behavioral experiments to show that T3 neurons respond omnidirectionally to the same visual stimuli that elicit bar tracking saccades, silencing T3 reduced the frequency of tracking saccades, and optogenetic manipulation of T3 acted on the saccade rate in a push–pull manner. Manipulating T3 did not affect smooth optomotor responses to large-field motion. Our results show that parallel neural pathways coordinate smooth gaze stabilization and saccadic bar tracking behavior during flight.

## Editor's evaluation

This study presents valuable new insights into the neural circuits in the fly visual system that mediate object tracking during flight, which have received less prior attention than the circuits involved in motion vision. The evidence supporting the claims of the authors is convincing. The work will be of interest to neuroscientists working on visual neural circuits and visually-guided behavior.

**\*For correspondence:**
frye@ucla.edu

**Competing interest:** The authors declare that no competing interests exist.

## Introduction

Discriminating and tracking a moving visual object against a cluttered and moving visual background represents a complex task that is nevertheless common across taxa. Flies have shown a stunning ability to actively orient toward and fixate foreground object stimuli, specifically long vertical bars, using detection algorithms that rely on few parameters and are remarkably robust to perturbation (*Egelhaaf, 1985*; *Reichardt et al., 1983*; *Theobald et al., 2008*). Bar orientation behavior becomes more challenging during locomotion when the visual scene moves across the retina generating complex patterns of optic flow that can move with or opposite the direction of the pursued bar (*Fox et al., 2014*). *Drosophila* has been a productive model for understanding how patterns of large-field optic flow are decoded by an array of small-field local motion detectors (*Borst, 2014*; *Mauss and Borst, 2020*) and how local motion direction is computed (*Borst et al., 2020*; *Groschner et al., 2022*; *Gruntman et al., 2019*). Small-field sampling of the visual world culminates in parallel ON and OFF directional motion detectors T4 and T5, respectively (*Joesch et al., 2010*; *Strother et al., 2017*). These two columnar neuron classes each have four subtypes tuned to a singular cardinal direction, which innervate one of four direction-selective layers of the lobula plate, the third neuropil of the optic lobe (*Fisher et al., 2015*; *Maisak et al., 2013*). Interneurons of the lobula plate pool inputs from

T4/T5 cells to assemble complex spatial filters for patterns of optic flow and project their axons to pre-motor descending pathways that have been proposed to coordinate directional head and wing steering movements to stabilize gaze against perturbations during locomotion (*Busch et al., 2018*; *Haikala et al., 2013*; *Heisenberg et al., 1978*). A motion opponency mechanism uses a group of lobula plate inhibitory interneurons to attenuate visual signals unrelated to self-motion (*Mauss et al., 2015*). Hence, the directional motion vision system is well suited to control optomotor gaze stabilization. Yet, the cellular mechanisms for coordinating bar tracking (i.e., exafferent motion) remain poorly understood.

Identifying foreground objects from a cluttered background requires some visual computations that either are not supported by or do not require the directional motion vision system (*Aptekar and Frye, 2013*; *Reichardt et al., 1983*). Several studies show that silencing directional motion detectors has little impact on object orientation behavior. For example, in tethered walking flies, synaptic suppression of T4/T5 cells dramatically reduces responses to bar motion but leaves orientation responses toward a stationary flickering bar largely unaffected (*Bahl et al., 2013*). In tethered flying flies, revolving a textured vertical bar around a textured circular arena generates a distinct initial steering response oriented counter directional to the moving bar, followed by a secondary response in the same direction of bar motion (*Keleş et al., 2018*). Silencing T4/T5 under these conditions reduced the secondary syn-directional tracking response, but left the initial counter-directional orientation phase intact (*Keleş et al., 2018*). These results would imply that a separate direction-independent subsystem that detects spatial contrast properties rather than motion direction per se might play a role in object orientation. Peripheral circuits have been shown to encode the requisite spatial contrast information (*Bahl et al., 2015*; *Bahl et al., 2013*).

The distinction between directional optomotor gaze stabilization and bar pursuit is accentuated by a behavioral paradigm in which a fly is tethered to a frictionless magnetic pivot and free to steer in the yaw plane. Under these conditions, yaw-free flies execute rapid body saccades to track a revolving narrow vertical bar. By contrast, they execute smooth steering movements to stabilize a revolving large-field panorama; smooth optomotor movement is interspersed with occasional nystagmus saccades or syn-directional 'catch up' saccades (i.e., optomotor saccades). But the dynamics of bar tracking saccades are distinct from large-field optomotor saccades (*Mongeau and Frye, 2017*). Perturbation of the directional motion vision pathway attenuates smooth optomotor responses, whereas perturbation of the spatial contrast pathway would be expected to compromise bar tracking saccades.

Here, we investigate a candidate pathway for object tracking by focusing on the response to elongated vertical bars that have been shown to provoke attractive orientation behavior in flying flies because they resemble plant stalks on which to land (*Maimon et al., 2008*; *Mongeau et al., 2019*; *van Breugel and Dickinson, 2012*). We confirm previous lines of evidence that T4/T5 coordinate smooth large-field optomotor responses that stabilize visual gaze and go substantially further to show that a parallel neural pathway, comprising T3 neurons, controls bar tracking saccades. T3 neurons arborize within single columns of the medulla and send axons into layers 2 and 3 of the fourth neuropil of the fly optic lobe, the lobula (*Fischbach and Dittrich, 1989*; *Takemura et al., 2013*). These neurons have previously been shown to respond to either small square objects or long vertical bars (*Keleş et al., 2020*; *Tanaka and Clark, 2020*). Using calcium imaging, we demonstrate that T3 neurons respond vigorously to the camouflaged motion-defined bars that robustly elicit tracking saccades (*Mongeau et al., 2019*; *Mongeau and Frye, 2017*). In rigidly tethered flies, hyperpolarizing T3 by genetically expressing an inward rectifier potassium channel (Kir2.1) reduces the initial counter-directional orientation response typically deployed for tracking motion-defined bars. In magnetically tethered flies, hyperpolarizing T3 neurons reduces the frequency of bar tracking saccades, whereas ON-OFF optogenetic activation by CsChrimson modulates a push–pull mechanism on saccade frequency. Finally, based on T3's physiological properties and the behavioral dynamics of bar tracking (*Mongeau and Frye, 2017*), we propose a model for how ensemble T3 activity can trigger bar tracking saccades through a behaviorally relevant integrate-and-fire mechanism (*Mongeau and Frye, 2017*).

## Results

### T3 neurons encode motion-defined bar movements

The lobula is heavily innervated by a class of visual projection neurons (VPNs), the lobula columnar (LC) cells, individuals of which, like olfactory receptor neurons, project together to the central brain forming bundles of type-specific terminals called optic glomeruli (*Aptekar et al., 2015*; *Panser et al., 2016*; *Wu et al., 2016*). In previous work (*Keleş et al., 2020*), our lab used an intersectional strategy to generate specific split-Gal4 driver lines for two T-shaped neuron types (*Pfeiffer et al., 2010*), T3 and T2a, arborizing in the medulla and terminating in layers 2 and 3 of the lobula (*Fischbach and Dittrich, 1989*; *Takemura et al., 2013*). The cell bodies of T3 and T2a are located caudally in the space between the medulla and lobula plate neuropiles (*Fischbach and Dittrich, 1989*).

To characterize responses to vertical bars by T3 neurons, we recorded calcium signals under in vivo two-photon excitation imaging with an LED-based visual stimulus (*Reiser and Dickinson, 2008*; *Figure 1A*). We imaged from a single presynaptic terminal in the lobula of flies expressing GCaMP6f in T3 (*Figure 1B*). Flies were presented with bars of varying directions and contrast polarities moving across the right visual field, ipsilateral to the recording site (*Figure 1C*). T3 showed an increase in calcium activity within individual presynaptic terminals, and the ensemble of T3 dendrites innervating layer 9 of the medulla exhibited a robust retinotopic wave of activation as the bar swept across the retina (*Figure 1C*, *Video 1*). Broadly consistent with prior results (*Keleş et al., 2020*), T3 neurons were strongly activated by front-to-back and back-to-front motion of either ON (brighter than background) or OFF (darker than background) bars, with a modest preference for OFF transitions (*Figure 1D and J*).

We measured the receptive field (RF) size by imaging from the presynaptic process of an individual T3 neuron (*Figure 1E*). We divided the right hemifield of the visual display into 10 azimuthal and eight elevation rectangular sampling bins and presented flies with a 2.25° dark bar moving within each bin in orthogonal directions (see 'Methods'). Responses to vertical and horizontal bar displacements were then multiplied to obtain the outer product (*Figure 1F*). To average RF size across flies, we selected the peak values of each bin and normalized them to the maximum value of the outer product. Finally, we spatially centered and averaged the RFs (*Figure 1G*). Average T3 RF size was mainly confined to the center 9° bin with almost no activity outside of it, which closely matched previous measurements (*Tanaka and Clark, 2020*). Confident that our recordings were being made from an individual T3, we tested the directional selectivity by comparing responses to dark bar movement in four cardinal directions. We found that T3 neurons were almost identically sensitive to leftward (back-to-front) and rightward (front-to-back) moving bars, as well as to upward and downward movements through the center of the RF, although with a higher variability to the latter (*Figure 1H*).

Next, we explored how T3 responded to patterned 'motion-defined' bars that robustly evoke tracking saccades (*Mongeau et al., 2019*; *Mongeau and Frye, 2017*). A motion-defined bar is composed of the same random ON/OFF pattern as the stationary background, and therefore is only detectable while in motion, by contrast to a classical luminance defined bar, which is brighter or darker than the surroundings and thus detectable even when stationary. Note that T4/T5 respond more strongly to a solid luminance-defined bar by comparison to a motion-defined bar (*Keleş et al., 2018*) due to sensitivity for longer spatial wavelength stimuli (*Agrochao et al., 2020*; *Groschner et al., 2022*; *Gruntman et al., 2019*; *Shinomiya et al., 2019*). T3 show rapid full-wave rectified response kinetics to flicker (*Keleş et al., 2020*). For this reason, these cells should be robustly excited by the high contrast frequency generated by motion-defined bars. As expected, T3 neurons responded strongly to these stimuli by comparison to solid ON and OFF bars (*Figure 1I and J*). As bar speed increased, the responses decreased both for motion-defined (*Figure 1I*) and solid OFF and ON bars (*Figure 1—figure supplement 1*).

As we did for T3, we characterized the responses to bar stimuli by T2a neurons, which innervate layers 1, 2, and 9 of the medulla (*Figure 1—figure supplement 2A*). Similar to T3, T2a has a small RF and omnidirectional sensitivity (*Figure 1—figure supplement 2B–E*). T2a showed strong responses to luminance-defined bars (with a preference for ON transitions) moving at a low speed (18° s$^{-1}$), but not at behaviorally relevant higher speeds (*Figure 1—figure supplement 2F*). Moreover, T2a showed weak-to-no responses to motion-defined bars (*Figure 1—figure supplement 2G*) and large-field gratings (*Figure 1—figure supplement 2H*). These results emphasize the distinct visual RF properties among different classes of columnar T-neurons and distinguish potential behavioral relevance

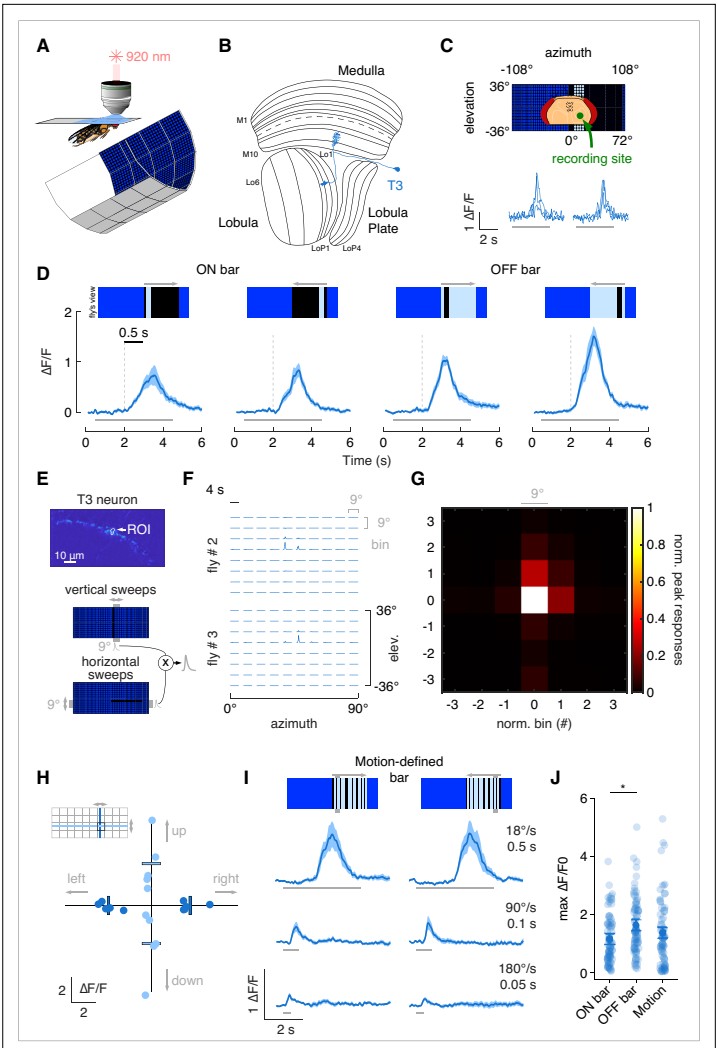

**Figure 1.** T3s are omnidirectional neurons with small receptive fields (RFs) and broad temporal sensitivity.
(**A**) Head-fixed fly for two-photon calcium imaging while presented with visual stimuli from a surrounding LED display. (**B**) Fly optic lobe neuropils (medulla, lobula, and lobula plate) with a T3 neuron highlighted in blue. (**C**) Top: schematic representation of the fly head and the recording site framed in the center of the LED display. Bottom: calcium imaging responses to an ON solid moving bar at 18° s⁻¹ (left: front-to-back; right: back-to-front) of a T3 neuron from a representative fly (three repetitions). (**D**) Average responses (mean ± SEM) to moving ON and OFF solid bars (9° × 72°, width × height) at 18° s⁻¹ in two different directions (front-to-back and back-to-front). Visual stimuli are depicted at the top. Dashed vertical gray lines indicate the coarse onset of the responses. Light gray horizontal bars at the bottom indicate stimulus presentation (n = 11 flies, three repetitions per fly). (**E**) Top: region of interest (ROI) drawn around the presynaptic terminal in the lobula of a single T3 neuron expressing GCaMP6f. Image representing the mean fluorescence from the two-photon imaging over the entire experiment in a representative fly. Bottom: representation of the procedure used to compute the RF of a single T3 neuron recording from its presynaptic terminal. Gray shaded region behind the LED display represents a bin (9° × 72°, width × height) within which a single pixel dark bar (2.25° width) is swept in two different directions. Calcium responses to vertical and horizontal sweeps are multiplied together. (**F**) Matrix of the multiplied traces in the 10 × 8 bins (horizontal × vertical) in two representative flies. The RF is probed within a window of 90° × 72° (horizontal × vertical). (**G**) Mean of the normalized peak responses by spatial location in individual T3 neurons (n = 5 flies, one repetition per fly). Bin = 0 represents the center of the RF. (**H**) Directional calcium peak responses to a 2.25° dark bar moving (18° s⁻¹) in the four cardinal directions of individual flies. Bars indicate the mean. Colors depict motion along different axes (horizontal: deep blue; vertical: light blue). (**I**) Average responses (mean ± SEM) to motion-defined bars (9° × 72°, width × height) moving in two different directions (front-to-back and back-to-front) at three different speeds (times on the right indicate how long it takes from the leading to the trailing edges). Light gray horizontal bars at the bottom indicate stimulus presentation (n = 11 flies, three repetitions per fly). (**J**) Maximum

*Figure 1 continued on next page*

*Figure 1 continued*

amplitude of the responses to ON, OFF, and motion-defined bars moving at 18° s$^{-1}$ (front-to-back and back-to-front responses are pulled together). Dark dots indicate the mean, the horizontal bars indicate SEM, and light dots represent peak amplitudes in single recordings. A linear mixed model was used to fit the data and ANOVA with pairwise post-hoc comparisons using *t*-tests adjusted with Bonferroni method were used to compare the predictions related to the types of bar (ON vs. OFF, p=0.01, Cohen's *d* = –0.51; ON vs. motion-defined: p=0.56, Cohen's *d* = –.023; OFF vs. motion-defined: p=0.33, Cohen's *d* = 0.28).

The online version of this article includes the following figure supplement(s) for figure 1:

**Figure supplement 1.** Speed sensitivity of T3 neurons.

**Figure supplement 2.** T2a neurons do not show a broad temporal sensitivity.

---

of T3 neurons to the bar tracking behaviors of flies. We therefore focus our behavioral analysis on T3 neurons.

## T3 neurons are tuned to speed rather than temporal frequency

We coarsely assessed T3 large-field responses by presenting gratings of two different spatial frequencies moving at different speeds in two directions. Responses to unilateral large-field gratings were nonselective for motion direction (*Figure 2A*). Stimulating with a unilateral large-field grating, by comparison to the bilateral gratings used in prior work, suggests that T3 neurons are not strictly object selective, even though center-surround mechanisms shape T3 RF properties (*Keleş et al., 2020*; *Tanaka and Clark, 2020*). Responses to an $\lambda$ = 18° grating moving at 2 Hz (36° s$^{-1}$) were similar to the responses to an $\lambda$ = 36° grating moving at 1 Hz (36° s$^{-1}$), suggesting that response amplitude was tuned to stimulus speed (*Figure 2B*). Similarly, the two spatial patterns presented at 90° s$^{-1}$ produced identical amplitude responses (*Figure 2C*). Unlike T3, which appear to be selective for pattern speed independently of spatial wavelength, T4/T5 neurons are tuned to the ratio of speed to spatial wavelength, temporal frequency (TF-tuned), rather than to the true stimulus speed (speed-tuned).

To more fully explore speed-tuning in T3 neurons, we tested grating patterns of six different spatial wavelengths, and six different temporal frequencies (TF) moving in two directions (i.e., front-to-back and back-to-front). Neurons that respond to visual motion can be categorized as TF- or speed-tuned (*Creamer et al., 2018*). In principle, TF-tuned neurons do not distinguish a small wavelength stimulus moving slowly from a long wavelength grating moving fast, as long as the temporal frequency is matched (*Figure 2D*). Instead, speed-tuned neurons track the ratio of these quantities because speed is the product of spatial wavelength and temporal frequency of the pattern (*Figure 2D*). Accordingly, TF-tuned neurons would show responses peaking at the same temporal frequency regardless of the spatial frequency, whereas speed-tuned neurons would show responses peaking at different temporal frequencies depending on the spatial frequency (*Figure 2E*).

The responses to different spatiotemporal gratings recorded from the presynaptic terminal of single T3 neurons showed neither a temporal nor a spatial frequency preference but a slanted cloud encircling a wide range of temporo-spatial conditions (*Figure 2—figure supplement 1*). To better visualize in the spatiotemporal frequency domain if T3 responses were TF-tuned with maximal peaks aligned horizontally (around a specific TF) or speed-tuned with maximal peaks aligned obliquely (around a specific speed), we fitted a regression line to the data representing the TF of the maximal responses per spatial frequency (*Figure 2F*). The slope of the regressions was consistent with speed-tuned neurons. Responses to gratings with low spatial wavelength evoked peak responses at higher TFs, whereas gratings with high spatial wavelength evoked peak responses at lower TFs (*Figure 2G*). Testing different models per fly with or without a regression coefficient representing speed-tuning or

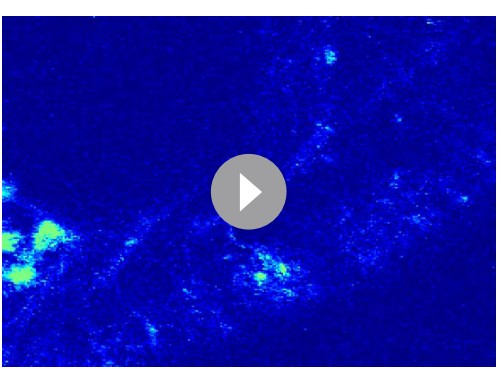

**Video 1.** Calcium imaging recording from T3 dendrites. T3 neurons expressing GCaMP6f respond to a back-to-front moving bright bar at 90° s$^{-1}$.
https://elifesciences.org/articles/83656/figures#video1

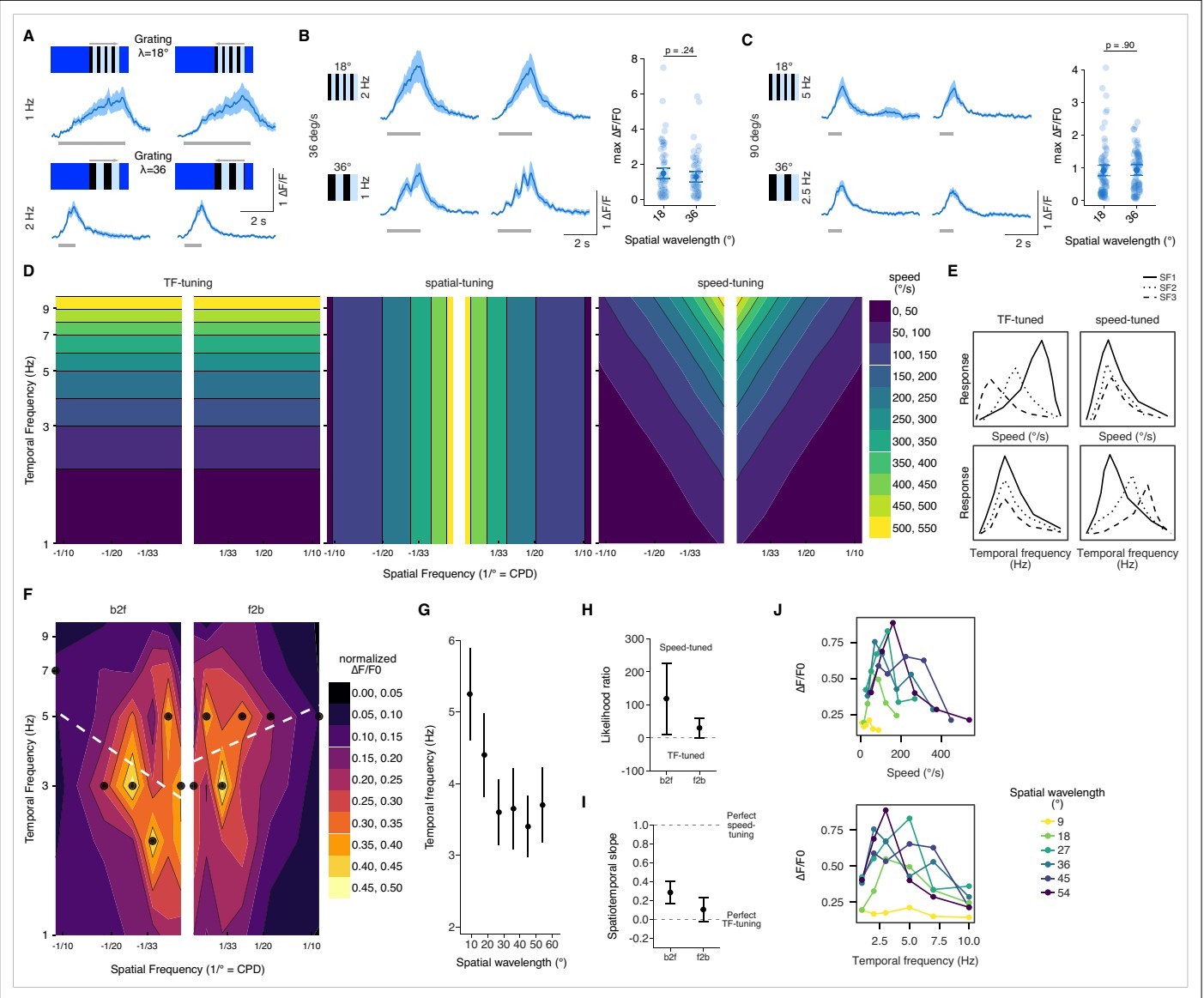

**Figure 2.** Speed tuning of T3 neurons. (**A**) Average responses of T3 to moving gratings of different spatial frequency (SF) and temporal frequency (TF) (n = 11 flies, three repetitions per fly). (**B**) Left: T3 neurons show similar responses for gratings moving at 36° s⁻¹ regardless of the SF of the stimuli ($\lambda$ = 18°, top; $\lambda$ = 36°, bottom). Right: maximum amplitude of the responses to gratings of different spatial wavelengths (18° and 36°) moving at 36° s⁻¹ (front-to-back and back-to-front responses are pulled together). Dark dots indicate the mean, the horizontal bars indicate SEM, and light dots represent peak amplitudes in single recordings. A linear mixed model was used to fit the data and a *t*-test was used to compare the predictions related to the different spatial wavelengths (18° vs. 36°, p=0.24, Cohen's *d* = 0.21). (**C**) Left: T3 neurons show similar responses for gratings moving at 90° s⁻¹ regardless of the SF of the stimuli ($\lambda$ = 18°, top; $\lambda$ = 36°, bottom). Right: as in (**B**), maximum amplitude of the responses to gratings of different spatial wavelengths moving at 90° s⁻¹ (18° vs. 36°, p=0.90, Cohen's *d* = –0.02). (**D**) Left: theoretical spatiotemporal plot on a log-log scale of a model tuned to the TF of the visual patterns (TF is in Hz and SF in cycle per degree [CPD]). Colored bands represent equal TFs. Middle: theoretical spatiotemporal plot of a spatial tuned model. Colored bands represent equal spatial frequencies. Right: theoretical spatiotemporal plot of a speed-tuned model. Colored bands represent equal speeds. (**E**) Hypothetical tuning curves of a visual neuron where the y-axis represents the calcium response. Different types of lines represent different SFs. Left column: TF-tuned neuron. At the top, speed-tuning curves of a TF-tuned neuron. At the bottom, temporal-tuning curves of the same neuron. Right column: speed-tuned neuron. As for the left column, top and bottom plots represent speed- and temporal-tuning curves, respectively. (**F**) Contour plots for the combinations of moving gratings based on the median of the normalized peak amplitudes per fly (n = 10 flies, one repetition per fly). Black dots indicate the TFs of the maximal normalized response for each SF. White dashed lines represent the regression lines drawn through the black dots. Thin black iso-response lines are plotted every 0.1 normalized ΔF/F0. Both axes are on a log scale. (**G**) Average TF (mean ± SEM) of the maximal normalized peak amplitudes per spatial wavelength (front-to-back and back-to-front responses are pulled together). TF plotted against spatial wavelength yields a line with slope equal to 0 if the response is TF-tuned, while the relationship between TF and spatial wavelength is inversely proportional if the response is speed-tuned. (**H**) Likelihood ratio of speed-tuned to TF-tuned models of T3 responses for back-to-front and front-to-back

*Figure 2 continued*

moving gratings. The dots mark the mean across flies of an approximation of the Bayes factor and the error bars indicate SEM. Positive values mark that the speed-tuned model is more likely than the TF-tuned model, whereas negative values mark that the TF-tuned model is more likely than the speed-tuned model. (**I**) Spatiotemporal slope of T3 responses. The dots represent the mean of the regression coefficients across flies as in (**F**) and the error bars are SEM. The thresholds depicted as dashed lines indicate: 1 = perfect speed-tuning; 0 = perfect TF-tuning. (**J**) Top: speed-tuning curves of T3 neurons. Dots represent the median of the maximal calcium responses for different speeds. Colors of the lines indicate different spatial wavelengths. Bottom: TF-tuning curves of T3. Dots indicate the median of the maximal calcium responses for different TFs. Spatial wavelengths are colored as at the top. T3 neurons show a selectivity for speeds between 100° s$^{-1}$ and 200° s$^{-1}$ regardless of spatial wavelengths, whereas the TFs are more spread out depending on the spatial wavelengths.

The online version of this article includes the following figure supplement(s) for figure 2:

**Figure supplement 1.** Spatiotemporal responses of T3 neurons.

TF-tuning model, respectively, and comparing them with an approximation of the Bayes factor (*Raftery, 1995*), we confirmed a higher probability in favor of a speed-tuned model (*Figure 2H*). However, the slopes of these models do not indicate perfect speed-tuning (*Figure 2I*). It is worth noting the break-point in this function – spatial wavelengths greater than 30° – produce TF-tuned responses, whereas gratings less than 30° produce speed-tuned responses (*Figure 2G*). Notwithstanding this variability in wavelength sensitivity, and regardless of the spatial wavelength, the responses peak closer to a fixed speed than a fixed temporal frequency (*Figure 2J*).

## T3 hyperpolarization reduces counter-directional bar orientation by rigidly tethered flies

Rigidly tethered flies respond to a bar revolving around a cylindrical visual display (*Figure 3A*) with a compound steering response composed of a counter-directional orientation component while the bar is in the visual periphery and a syn-directional tracking component while the bar crosses the visual midline (*Keleş et al., 2018*; *Reiser and Dickinson, 2010*). *Figure 3A* shows this compound response in schematic form. As the bar moves from the rear into the periphery, the fly initially steers toward the bar's position, opposite (negative) its direction of motion (*Figure 3A*, red trace). As the bar moves towards visual midline, the steering effort switches to track (positive) the direction of bar motion (blue trace). The initial counter-directional and following syn-directional turns sum to zero net steering effort once the bar is on visual midline – the fly is oriented directly at the bar.

Flies expressing inwardly rectifying potassium channel Kir2.1 (*Baines et al., 2001*) in T4/T5 neurons showed weakened syn-directional tracking responses, but intact counter-directional orientation responses (*Keleş et al., 2018*; *Figure 3B*). We repeated this experiment after silencing T3 neurons and we compared it with the one from control flies. As controls, we opted for flies using exactly the same driver and genomic insertion sites, with the only difference that the control driver lacks the regulatory fragment (also known as enhancer) that leads to effector expression (i.e., enhancerless split-Gal4 driver, also known as empty split-Gal4). This genotype represents in our opinion the best control for our behavioral experiments since the genetic background matches the experimental flies (differing only for the *cis*-regulatory modules), and it has been validated by our and other labs (*Ache et al., 2019*; *Klapoetke et al., 2017*; *Namiki et al., 2022*).

Compared to the responses of empty split-Gal4 crossed with UAS-Kir2.1, the counter-directional orientation responses of T3 silenced flies were significantly reduced, whereas T4/T5 silenced flies were unaffected (*Figure 3B*, left). On average, flies with hyperpolarized T3 neurons responded to the revolving bar by always steering (ΔWBA) in the direction of motion. Thus, as the bar crossed visual midline, these animals seemed to 'anticipate' the bar (ΔWBA > 0 when the bar was at visual midline). By contrast, controls and T4/T5 silenced flies showed an initial counter-directional orientation response, followed by a weakened syn-directional response so that steering is balanced (ΔWBA = 0) as the bar crossed midline (*Figure 3B*, left). Quantification of the steering effort as the bar crossed midline indicates significant influence of hyperpolarizing T3, but not T4/T5 or the genetic control (*Figure 3B*, right). This phenotype was not biased by off-target expression because our T3 split-Gal4 driver line showed no cellular labeling elsewhere in the brain or ventral nerve cord (VNC) (*Figure 3— figure supplement 1*).

To better visualize the effects of hyperpolarizing T3, we integrated ΔWBA over time as a measure of the fictive fly angular displacement (i.e., integral of angular velocity) as a measure of the accumulating

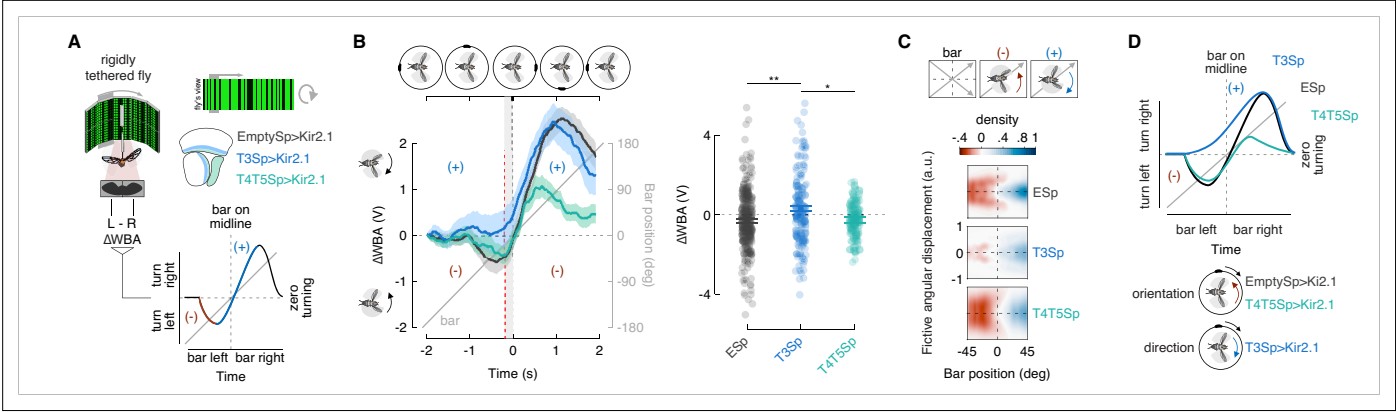

**Figure 3.** Constitutive silencing of T3 compromises the orientation response. (**A**) Left: cartoon of the rigid-tether setup in which a fly is glued to a tungsten pin and placed within a surrounding LED display presenting a random pattern of bright and dark stripes. An infrared diode above the fly casts a shadow on an optical sensor that records the difference between the left and right wing beat amplitudes (ΔWBA). Right: schematic representation of the optic lobe regions where Kir2.1 channels were expressed in the three genotypes tested (data referred to T4/T5Sp>Kir2.1 flies are reproduced from *Keleş et al., 2018*). Bottom: schematic diagram of the experiment. A bar revolves around the fly (gray). Initially, steering is in the direction opposite to bar motion (plotted in red, negative quadrant), followed by steering in the same direction as the bar (plotted in blue, positive quadrant). Depending on the strength of each response, the steering effort may be non-zero when the bar is at zero degrees (midline). (**B**) Top: schematic of the bar positions over time. Left: population average time-series steering responses (mean ± SEM) in the three genotypes tested (T4/T5Sp data replotted from *Keleş et al., 2018*) to a motion-defined bar revolving at 90° s⁻¹ (responses to counterclockwise [CCW] rotations were reflected and pooled with clockwise [CW] responses). Gray shaded region (between the vertical red and black dashed lines) represents a 200 ms time window before the bar crosses the fly's visual midline (n = 44 EmptySp>Kir2.1, n = 26 T3Sp>Kir2.1, n = 22 T4/T5Sp>Kir2.1). Note that T3Sp>Kir2.1 reduces counter-directional steering, whereas T4/T5Sp>Kir2.1 reduces syn-directional steering. Right: dot plot average ΔWBA values across the 200 ms time window per trial. Dark dots indicate the mean, horizontal bars indicate SEM, and light dots represent the values in single recordings. A linear mixed model was used to fit the data and ANOVA with pairwise post-hoc comparisons using *t*-tests adjusted with Bonferroni method were used to compare the three genotypes (F(2, 89) = 5.83, p=0.004; EmptySp vs. T3Sp, p=0.005, Cohen's *d* = –0.40;; EmptySp vs. T4/T5Sp, p=1, Cohen's *d* = 0.03; T3Sp vs. T4/T5Sp, p=0.03, Cohen's *d* = 0.38). (**C**) Heatmaps of flies' steering effort at the population level in the three genotypes as function of the fictive fly angular displacement (arbitrary unit) and bar position (data are mirrored along the x-axis in order to get a uniform directional distribution). EmptySp and T4/T5Sp show a strong counter-directional response (red blob) while T3Sp show only a very slight counter-directional response. (**D**) Top: schematic summary of experimental results. T4/T5Sp>Kir2.1 reduces the syn-directional tracking effort while leaving the counter-directional (-) orientation response intact. T3Sp>Kir2.1 reduces the counter-directional steering effort, leaving the syn-directional response (+) intact, and therefore steers leftward of the bar as it crosses midline. Bottom: cartoon depicting normal orientation response in T4/T5Sp>Kir2.1 flies and compromised in T3Sp>Kir2.1 flies.

The online version of this article includes the following figure supplement(s) for figure 3:

**Figure supplement 1.** Expression pattern of T3 neurons.

**Figure supplement 2.** Hyperpolarization of T3 does not compromise the syn-directional response.

**Figure supplement 3.** Silencing of T3 does not affect frontal bar fixation in rigid closed-loop.

strength of the optomotor response, or 'wind-up.' For each fly, we zoomed in on a time window (1 s) around the bar crossing the visual midline (time = 0), stretched the traces representing the angular displacement to have a common frame of reference across flies, and color-coded the distribution of traces based on direction and magnitude of ΔWBA normalized to the maximal absolute value per fly. Reflecting, pooling traces to the two directions of bar revolution and smoothing with a Gaussian produced a density heatmap of the steering effort as function of the fictive fly angular displacement and bar position (given that time and bar position are correlated). These plots reveal reduced counter-directional orientation responses (red shading) for T3 silenced flies by comparison with controls and T4/T5 silenced flies (*Figure 3C*). These plots visually highlight the effect evident within the average ΔWBA time series (*Figure 3B*).

For bar stimuli that evoke significantly stronger syn-directional steering effort, such as a 30° solid dark bar, silencing T3 had no effect, whereas, as shown previously (*Keleş et al., 2018*), blocking T4/T5 neurons did (*Figure 3—figure supplement 2A and B*). Confirming the results of *Figure 3*, the typical advanced response generated by winding up the optomotor system, which is dependent on T4/T5 neurons, was intact in T3 silenced flies (*Figure 3—figure supplement 2C and D*). The early small counter-directional response that both control and T3 silenced flies show is likely driven by T4/

T5 neurons through parallel pathways (e.g., LPLC1 neurons). Synaptic silencing of T4/T5 reduced the turning toward small dark objects moving back-to-front (*Tanaka and Clark, 2022*). Noteworthy in our data is the disappearance of the early counter-directional response component in T4/T5 silenced flies, but the emergence of a later counter-directional component – similar to the motion-defined bar (*Figure 3—figure supplement 2B and D*). This effect may be due to an imbalance of T4/T5 silencing to favor the steering responses driven by intact T3 activity. Our interpretation is that when the directional computation is challenged with high-frequency motion-defined bars, then the influence of T3 neurons is exaggerated, resulting in stronger counter-directional orientation. On the contrary, luminance-defined bars drive the directional motion vision pathway (i.e., T4/T5) more strongly than motion-defined bars (*Keleş et al., 2018*), overcoming the counter-directional response of T3. In T4/T5 silenced flies the syn-directional component is not completely eradicated. This might be due to T3 neurons taking over or to the residual functionality of T4/T5 since the silencing approach we adopted does not fully erase the activity of the cells. Moreover, virtual closed-loop bar fixation experiments have shown to depend on T4/T5 neurons (*Fenk et al., 2014*), but silencing T3 had no effect on this behavior regardless of the type of bar (*Figure 3—figure supplement 3*).

The parsimonious interpretation of these results is that, for rigidly tethered flies, the syn-directional optomotor tracking response to a moving bar is dependent upon T4/T5 activity, whereas the counter-directional orientational response is dependent upon T3 activity (*Figure 3D*).

## T3 hyperpolarization reduces saccadic bar tracking in magnetically tethered flies

In freely flying flies, orientation responses to visual objects trigger rapid body rotations called saccades, which are functionally analogous to our own gaze stabilizing rapid eye movements (*Land, 1992*; *Tammero and Dickinson, 2002*; *van Breugel and Dickinson, 2012*). Behavioral results and theoretical models suggest that orientation responses by rigidly tethered flies and saccadic tracking responses by magnetically tethered flies are coordinated by non-directional (or omnidirectional) positional feature detectors with sensitivity to the high-frequency transients generated by motion-defined bars, naturalistic stimuli that might not strongly activate directional motion detectors (*Keleş et al., 2018*; *Mongeau and Frye, 2017*; *Reichardt and Poggio, 1976*). T3 cells indeed show these physiological properties (*Figure 1 and 2*) and are required for intact counter-directional orientation steering effort (*Figure 3*). We therefore tested the functional role of T3 neurons for saccadic bar tracking by using magnetically tethered flies, free to steer in yaw on a frictionless pivot and execute robust body saccades (*Figure 4A*, top).

Following the approach of prior work (*Mongeau and Frye, 2017*), we elicited bouts of tracking saccades by revolving a motion-defined bar against a stationary background (*Figure 4A*, bottom). In magnetically tethered flies, saccades are easily identified by characteristic impulses in angular velocity resulting in stepwise changes in flight heading (*Figure 4A*, right, *Video 2*). We hyperpolarized T3 and T4/T5 neurons by expressing Kir2.1 channels with split-Gal4 lines (*Video 3*, *Video 4*). We first compared the angular displacement that flies traveled in response to bar motion. Each trial was parsed into 5 s bins, normalized for initial heading (0°). Assuming bilateral symmetry, we reflected the counterclockwise (CCW traces to represent responses to CW motion (*Figure 4B*, positive-going cumulative angle)). We spatially pooled the individual traces to generate a population heatmap of the angular displacement traveled within each 5 s epoch. We used as controls empty split-Gal4 crossed with UAS-Kir2.1 flies. By visual inspection, controls dispersed within the first second, but T3 silenced flies remained concentrated at their initial heading, indicating that they were not tracking the motion-defined bar (*Figure 4B*). T4/T5 silenced flies dispersed similar to empty vector controls indicating that they were able to follow to some extent. We next computed the frequency of body saccades during the trials, confirming the rate for our controls (~1 Hz) is in agreement with that of wild-type flies (*Mongeau and Frye, 2017*). Both T3 and T4/T5 silenced flies showed a lower frequency of saccades than controls, suggesting some role of both cell types (*Figure 4C*).

In order to gain further insight into this result, we computed the general performance index (PI) in the three genotypes tested – operationalized as the ratio between angular displacement traveled by the fly, and angular displacement traveled by the bar (*Figure 4D*). Control flies ended up with a positive PI in only ~50% of trials because they often tried to catch up with the bar with saccades oriented counter-directionally, resulting in negative PI values (*Figure 4—figure supplement 1*). T3 silenced

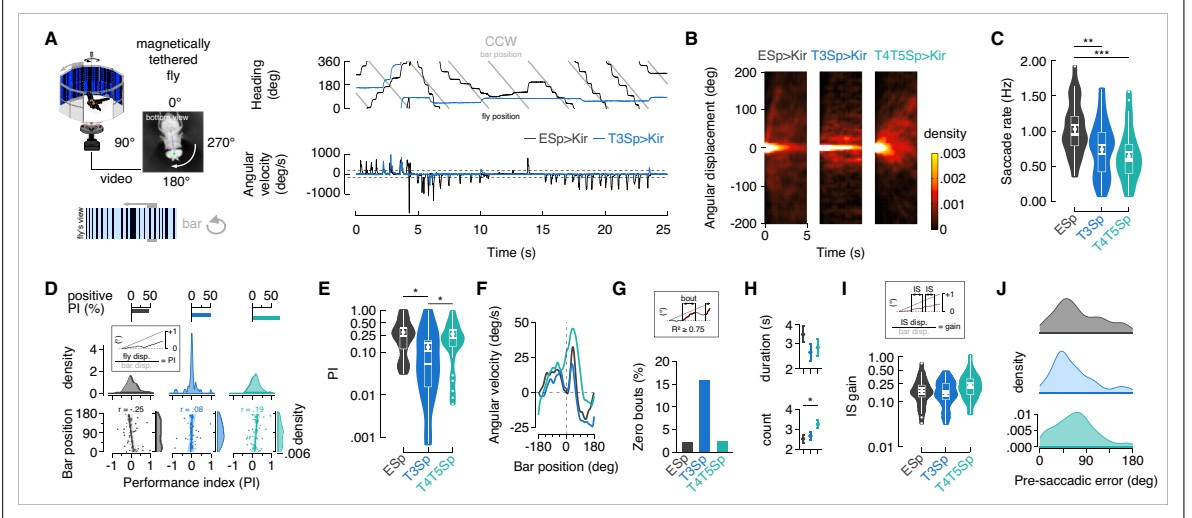

**Figure 4.** Flies with T3 hyperpolarized track a motion-defined bar poorly. (**A**) Top-left: cartoon of the magnetic-tether setup in which a fly is glued to a stainless steel pin and suspended within a magnetic field in turn placed within a surrounding LED display presenting a random pattern of bright and dark stripes. Infrared diodes illuminate the fly from below and a camera captures videos of the fly's behavior from the bottom. Bottom-left: motion-defined bar moving counterclockwise (CCW) from the fly's perspective. Top-right: wrapped heading traces from two representative flies (black: EmptySp>Kir2.1; blue: T3Sp>Kir2.1) responding to a CCW revolving motion-defined bar at 112.5° s⁻¹ for 25 s. Gray line represents the bar position (in this plot it relates to the EmptySp fly). Bottom-right: filtered angular velocity profiles referring to the two representative flies at the top. Horizontal dashed lines indicate the threshold for detecting saccades. (**B**) Heatmaps of the angular displacement (unwrapped) traveled by flies within bins of 5 s in the three genotypes during the rotation of a motion-defined bar (n = 23 EmptySp>Kir2.1, n = 22 T3Sp>Kir2.1, n = 22 T4/T5>Kir2.1). (**C**) Violin-box plots of the frequency of bar tracking saccades per trial in the three genotypes (black: EmptySp>Kir2.1, 1240 saccades; blue: T3Sp>Kir2.1, 851 saccades; green: T4T5Sp>Kir2.1, 708 saccades). Big white dot represents the mean, thin horizontal bars indicate SEM, and thick horizontal bar indicates the median. Small white dots on the violin tails represent outliers. A linear mixed model was used to fit the data and pairwise post-hoc comparisons using t-tests adjusted with Bonferroni method were used to compare the predictions (EmptySp vs. T3Sp, p=0.007, Cohen's d = 1.01; EmptySp vs. T4/T5Sp, p=0.0002, Cohen's d = 1.35; T3Sp vs. T4/T5Sp, p=0.87, Cohen's d = 0.34). (**D**) Top: bar plot representing the percentage of trials in which flies showed a positive performance index (PI) per genotype. Color-code as in (**C**). A PI was computed based on the ratio between angular displacement of the fly and angular displacement of the bar at the end of the trial (inset). Middle: kernel density estimation of the PI distribution in the three genotypes. T3Sp>Kir2.1 flies show a high peak around zero. Bottom: scatter plot of the PI plotted against the initial bar position (at frame = 1). A regression line was drawn across the data points (thick line) and a Pearson correlation coefficient (r) computed (EmptySp, p=0.09; T3Sp, p=0.59; T4/T5, p=0.22). On the right of each scatter plot, kernel density estimation of the initial bar position distribution. Colored shade around the regression line represents SEM. (**E**) Violin-box plots of the PI (y-axis is on a log scale) in trials with positive values (greater than 0). PI mean for clockwise (CW) and CCW revolving bars was computed and a generalized linear model (with gamma distribution and log link function) was fitted to the data (EmptySp vs. T3Sp, p=0.04, Cohen's d = 0.80; EmptySp vs. T4/T5Sp, p=1, Cohen's d = 0.09; T3Sp vs. T4/T5Sp p=0.05, Cohen's d = –0.71). Graph features are as in (**C**). (**F**) Average angular velocity as a function of the bar position per genotype. T4T5Sp>Kir2.1 flies steer syn-directionally to the bar even when it is at the visual midline. (**G**) Bouts of bar tracking behavior were defined as periods of tracking in which the R-squared captured by linear regression was equal or greater than 0.75 (inset). Bar plot of the percentage of trials where zero bar tracking bouts were detected. (**H**) Top: average duration (mean ± SEM) of bar tracking bouts across flies on a log scale (EmptySp vs. T3Sp, p=0.38, Cohen's d = 0.60; EmptySp vs. T4/T5Sp, p=0.44, Cohen's d = 0.51; T3Sp vs. T4/T5Sp, p=1, Cohen's d = –0.16). Bottom: average number (mean ± SEM) of bar tracking bouts per trial (EmptySp vs. T3Sp, p=1, Cohen's d = –0.15; EmptySp vs. T4/T5Sp, p=0.04, Cohen's d = –0.75; T3Sp vs. T4/T5Sp, p=0.17, Cohen's d = –0.60). (**I**) Inter-saccadic (IS) gain was computed as the ratio between angular displacement of the fly and angular displacement of the bar during IS intervals (inset). Violin-box plots of IS gain (EmptySp vs. T3Sp, p=1, Cohen's d = 0.27; EmptySp vs. T4/T5Sp, p=0.33, Cohen's d = –0.63; T3Sp vs. T4/T5Sp, p=0.08, Cohen's d = –0.90). Graph features and statistical approach are as in (**E**) with the only difference that a mixed-effects model was used. (**J**) Kernel density estimation of the pre-saccadic error angle distribution in the three genotypes. Color-code as in (**A**).

The online version of this article includes the following figure supplement(s) for figure 4:

**Figure supplement 1.** Raw unwrapped angular distance traveled by the flies during bar tracking.

**Figure supplement 2.** Saccade dynamics during bar tracking.

**Figure supplement 3.** T3 silencing does not affect the response to the rotation of a wide-field panorama.

**Figure supplement 4.** Raw unwrapped angular distance traveled by the flies during wide-field stabilization.

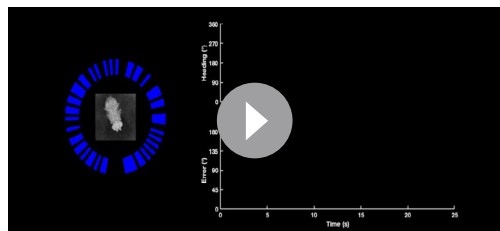

**Video 2.** EmpySp>Kir2.1 fly presented with a revolving motion-defined bar. Left: bottom view of a single fly within an animated cartoon of the surrounding display presenting a motion-defined bar revolving for 25 s at 112.5° s⁻¹. Right-top: fly heading (white) and bar position (gray). Right-bottom: error (red) between bar position and fly heading. EmptySp flies track the bar by using a saccade-and-fixation strategy. Original video recorded at 200 fps.

https://elifesciences.org/articles/83656/figures#video2

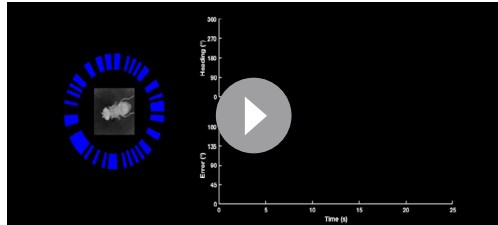

**Video 4.** T4/T5Sp>Kir2.1 fly presented with a revolving motion-defined bar. Conditions identical to *Video 2*. T4/T5Sp flies track the bar with some defects in gaze stabilization.

https://elifesciences.org/articles/83656/figures#video4

flies showed a similar overall percentage of positive PI as controls, but the distribution of PI was very different for T3 (*Figure 4D*, top, *Figure 4—figure supplement 1*). Genetic controls and T4/T5 hyperpolarized flies dispersed, generating a range of PI around the mean, whereas T3 hyperpolarized flies did not disperse, generating a high probability of PI near zero (*Figure 4D*, middle). Curiously, T4/T5 showed more positive PI than controls (*Figure 4D*, top), indicating that these flies were more inclined to steer syn-directionally with the bar. We then wondered if the initial bar position (at frame = 1) relative to the fly's heading at the start of the trial could partially explain the similarity in the percentage of positive PI between controls and T3 silenced flies. Indeed, controls showed a negative correlation between initial bar position and resulting PI, meaning that when the bar starts from 180°, the flies tend to steer counter-directionally toward it (PI < 0), whereas, consistent with their small displacement response (*Figure 4B*), T3 silenced flies did not show any correlation between initial bar position and PI (*Figure 4D*, bottom). Instead, T4/T5 silenced flies showed a positive correlation, meaning that they tended to steer in the direction of bar motion (PI > 0) (*Figure 4D*, bottom). Taking into account only positive PI values, flies expressing Kir2.1 in T3 showed a clear compromise in bar tracking performance compared to controls (*Figure 4E*). Interestingly, flies expressing Kir2.1 in T4/T5 showed average PI comparable to the controls (*Figure 4E*), meaning that they were able to track the bar without performing as many saccades (*Figure 4C*).

T4/T5 silenced flies seem to respond to a moving bar by saccading less, like T3, but smoothly steering more. To delve further into this phenomenon, we measured the average angular velocity as a function of the bar position. Silencing T4/T5 produced an overall increase of angular velocity by comparison with controls, especially when the bar was near the visual midline (*Figure 4F*). We then counted bouts of bar tracking behavior – operationalized as sliding non-overlapping windows corresponding to half a revolution of the bar in which at least 75% of variance in the fly angular displacement was explained by the bar angular displacement (computed by fitting a linear regression). As expected, the percentage of trials without any bar tracking bouts was higher in T3 hyperpolarized flies but equal between control and T4/T5 hyperpolarized flies (*Figure 4G*). When bouts did occur, the average duration did not differ among genotypes, but their number of tracking bouts per trial

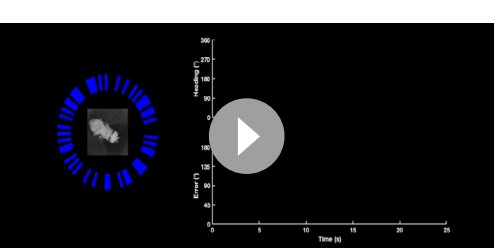

**Video 3.** T3Sp>Kir2.1 fly presented with a revolving motion-defined bar. Conditions identical to *Video 2*. T3Sp flies do not track the bar.

https://elifesciences.org/articles/83656/figures#video3

was higher in T4/T5 silenced flies than controls (*Figure 4H*). Finally, we analyzed the inter-saccadic gain (ratio of body rotation to stimulus rotation between saccades) during bar tracking bouts. The silencing of T4/T5 led tendentially to higher inter-saccade tracking gain than controls and T3 silenced flies (*Figure 4I*). Taken together, these results indicate that flies with silenced T4/T5 were 'pushed around' by the moving bar. This may seem counterintuitive since T4/T5 provide the directional motion signals for optomotor steering, so their blockade should be expected

to attenuate smooth bar responses. Consider, however, that smooth bar tracking behavior is a result of the inability to properly stabilize the visual panorama in T4/T5 silenced flies. Following the bar resulted in extended pre-saccadic error angles for T4/T5 silenced flies (*Figure 4J*). The corollary is that integrating the bar error to trigger a saccade (*Mongeau and Frye, 2017*) requires stable visual gaze. We also measured the angular velocity profile of saccades, which did not differ across the three genotypes (*Figure 4—figure supplement 2A*), and nor did saccade duration differ (*Figure 4—figure supplement 2B*). However, saccade amplitude was slightly reduced in both T3 and T4/T5 blocked flies (*Figure 4—figure supplement 2C*).

Silencing T4/T5 would be expected to compromise large-field optomotor gaze stabilization. We compared the silencing effects in response to rotation of a full large-field panorama, T4/T5 silenced flies showed higher percentage of trials with PI equal or less than zero (i.e., no stabilization), reduced positive PI value, and reduced smooth inter-saccadic gain (*Figure 4—figure supplement 3A–C*). These effects are in agreement with a compromised directional optomotor system (*Maisak et al., 2013*). The frequency of saccades was similar among the three genotypes, even if slightly reduced in T4/T5 silenced flies (*Figure 4—figure supplement 3D*). For large-field evoked saccade dynamics, subtle differences were evident in T3 and T4/T5 silenced flies compared to controls (*Figure 4—figure supplement 3E–G*). It is noteworthy that hyperpolarization of T3 neurons did not affect large-field elicited gaze stabilization in any measurable way (*Figure 4—figure supplement 4*).

## T3 depolarization suppresses saccadic bar tracking in magnetically tethered flies

Constitutive hyperpolarization of T3 shows that normal excitatory activity in these cells is required for saccadic bar tracking pursuit in magnetic tethered flight (*Figure 4*). We sought to further support this result with an inducible perturbation. We expressed CsChrimson (*Klapoetke et al., 2014*) channels in T3 and T4/T5 neurons, and stimulated intermittently with 5 s periods of red light ON (each one followed by 5 s of light OFF), while presenting flies with a revolving motion-defined bar (*Figure 5A*). This approach was 'non-selective' in the sense that it artificially activates the population of columnar neurons together, rather than in the spatially localized retinotopic manner they would be normally active. Therefore, we consider this approach to be a 'loss-of-function' perturbation, rather than a 'gain-of-function' excitation of T3, that is of course dynamic rather than static (e.g., Kir2.1). To generate control flies, we adopted the same method as the Kir2.1 silencing experiments by crossing empty split-Gal4 flies with the UAS effector flies (i.e., UAS-CsChrimson). Our aim was to assess how bar tracking behavior was impacted by both phases of the perturbation: light ON saturating population depolarization or light OFF recovery. In enhancerless controls, the light ON did provoke small changes in the flies' heading (*Figure 5B*, top black, *Figure 5—figure supplement 1*). Activation of T3 neurons at high LED intensity tended to reduce or eliminate active bar tracking (*Figure 5B*, middle blue, *Figure 5—figure supplement 1*). Population depolarization of T4/T5 seemed not to change saccadic tracking in any obvious way (*Figure 5B*, bottom green, *Figure 5—figure supplement 1*). To quantify the effects of dynamic depolarization, we computed the PI during the periods of light ON and OFF individually in the three genotypes (*Figure 5C*). The percentage of trials with PI > 0 (active bar tracking) was high in T3>CsChrimson flies when the light was OFF (*Figure 5C*), whereas the same flies shifted to zero PI (no bar tracking) when the light was ON (*Figure 5C*), recapitulating the effect of the silencing manipulation (*Figure 4D*). CsChrimson activation had no obvious influence over the distribution of PI for T4/T5.

To better visualize how bar tracking performance changed over time, we binned together the unwrapped heading traces segregated by OFF and ON epochs, normalized to the initial heading, and generated heatmaps of the resultant cumulative heading angle. We split the heatmaps into two spatial windows: a first window for 0–200° angular displacements from the animal's initial heading, and a second window ranging 200–500° of angular displacements from the initial fly's heading. For angular displacements falling within the first window, the heatmaps between CsChrimson OFF and ON epochs were not obviously different across the three genotypes (*Figure 5D*, lower). However, light-gated depolarization of T3 neurons caused cessation of bar responses before the first 360° revolution around the circular arena, whereas controls and T4/T5>CsChrimson flies, on average, continued to steer throughout the trial (*Figure 5D*, upper). T3>CsChrimson flies were unable to continually follow the bar. By comparing the positive PI values (greater than zero) between OFF and ON epochs in

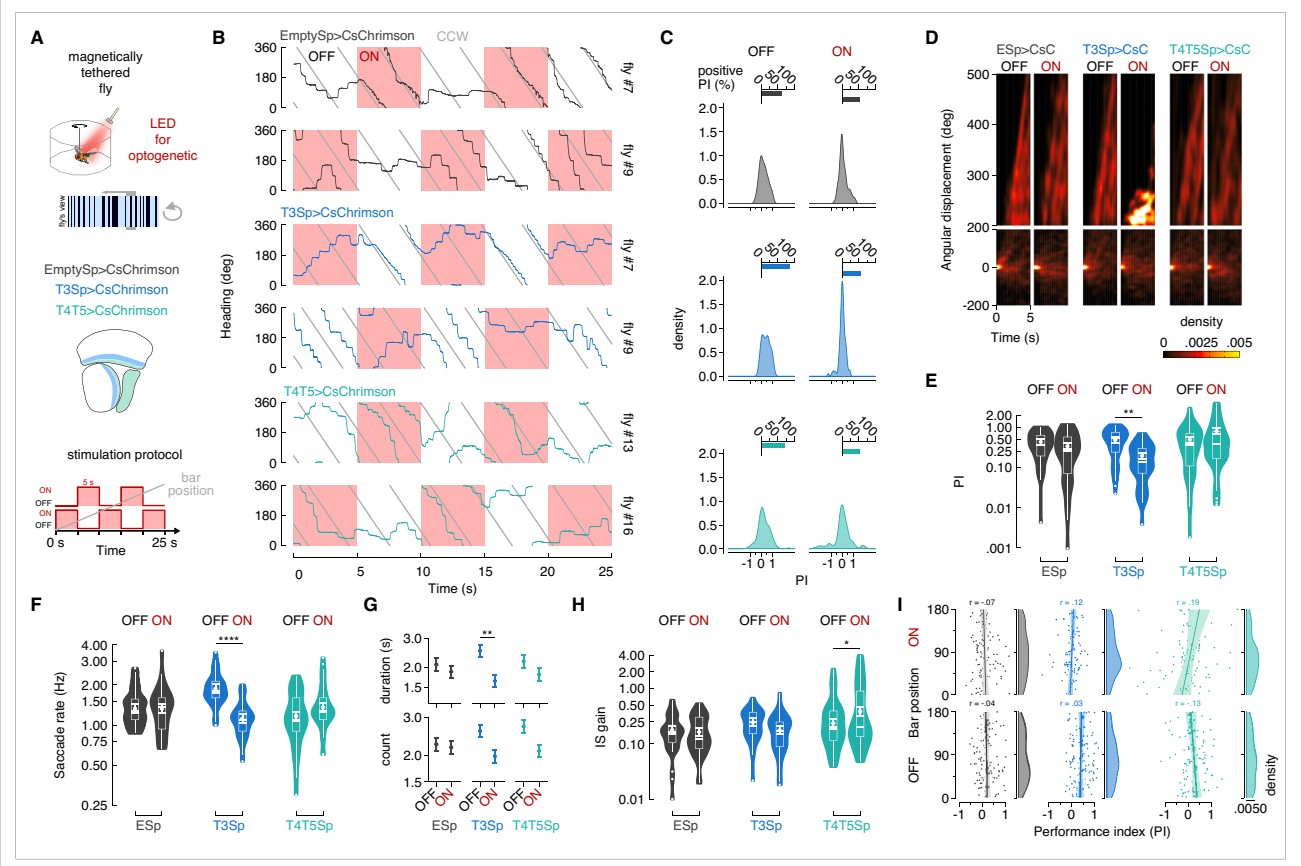

**Figure 5.** T3 decreases bar tracking behavior during non-selective depolarization. (**A**) Top: cartoon of the magnetic-tether setup implemented with an optogenetic LED for neural stimulation. Middle: genotypes tested and schematic representation of the optic lobe regions innervated by T3 and T4/T5. Bottom: schematic representation of the stimulation protocol: repetition of 5 s optogenetic LED OFF followed by 5 s LED ON for 25 s. ON/OFF starting was randomly selected. (**B**) Wrapped heading traces from six representative flies (black: EmptySp>CsChrimson; blue: T3Sp>CsChrimson; green: T4/T5Sp>CsChrimson) responding to a counterclockwise (CCW) revolving motion-defined bar as in *Figure 4A*. Red shaded regions represent periods of LED ON. EmptySp>CsChrimson and T4/T5EmptySp>CsChrimson flies were slightly affected by the LED ON, while T3Sp>CsChrimson flies stop chasing the bar. (**C**) Top: bar plot representing the percentage of trials in which flies showed a positive performance index (PI) during periods of LED OFF and ON in the three genotypes. A PI was computed as in *Figure 4D*. Color-code as in (**B**). Bottom: kernel density estimation of the PI distribution per genotype and LED condition. T3Sp>CsChrimson flies show a high peak around zero when the LED was ON. (**D**) Heatmaps of the angular displacement (unwrapped) traveled by flies within bins of 5 s in the three genotypes during the rotation of a motion-defined bar (n = 20 EmptySp>CsChrimson, n = 22 T3Sp>CsChrimson, n = 21 T4/T5>CsChrimson). The map was divided in two windows to highlight the behavior of flies traveling an angular displacement greater than 200°. T3Sp>CsChrimson flies remain stuck during the optogenetic stimulation periods. (**E**) Violin-box plots of the PI in trials with positive values (y-axis is on a log scale). Big white dot represents the mean, thin horizontal bars indicate SEM, and thick horizontal bar indicates the median. Small white dots on the violin tails represent outliers. PI mean for clockwise (CW) and CCW revolving bars was computed and a generalized linear model (with gamma distribution and log link function) was fitted to the data (EmptySp OFF vs. ON, p=1, Cohen's d = 0.33; T3Sp OFF vs. ON, p=0.002, Cohen's d = 1.21; T4/T5Sp OFF vs. ON, p=0.53, Cohen's d = –0.67). (**F**) Violin-box plots of the frequency of bar tracking saccades per trial during periods of LED OFF and ON (y-axis is on a log scale). A generalized linear mixed model was used to fit the data (residuals were fitted with a gamma distribution and a log link function) and pairwise post-hoc comparisons using t-tests adjusted with Bonferroni method were used to compare the predictions (EmptySp OFF vs. ON, p=1, Cohen's d = 0.01; T3Sp OFF vs. ON, p<0.0001, Cohen's d = 1.70; T4/T5Sp OFF vs. ON, p=0.24, Cohen's d = –0.52). Graph features are as in (**E**). (**G**) Bouts of bar tracking behavior (defined as in *Figure 4G*) during periods of LED OFF and ON. Top: average duration (mean ± SEM) of bar tracking bouts across flies on a log scale (EmptySp OFF vs. ON, p=1, Cohen's d = 0.26; T3Sp OFF vs. ON, p=0.007, Cohen's d = 1.09; T4/T5Sp OFF vs. ON, p=1, Cohen's d = 0.48). Bottom: average number (mean ± SEM) of bar tracking bouts during OFF or ON periods on a log scale (EmptySp OFF vs. ON, p=1, Cohen's d = 0.10; T3Sp OFF vs. ON, p=0.08, Cohen's d = 0.83; T4/T5Sp OFF vs. ON, p=0.13, Cohen's d = 0.81). (**H**) Violin-box plots of inter-saccadic (IS) gain (defined as in *Figure 4I*) during periods of LED OFF and ON (EmptySp OFF vs. ON, p=1, Cohen's d = 0.12; T3Sp OFF vs. ON, p=0.26, Cohen's d = 0.49; T4/T5Sp OFF vs. ON, p=0.01, Cohen's d = –0.75). Graph features and statistical approach are as in (**F**). (**I**) Scatter plot of the PI plotted against the bar position at the beginning of each 5 s period in ON and OFF conditions. A regression line was drawn across the data points (thick line) and a Pearson correlation coefficient (r) computed (EmptySp OFF, p=.068; EmptySp ON, p=0.48; T3Sp OFF, p=0.73; T3Sp ON, p=0.20; T4/T5Sp OFF, p=0.19; T4/T5Sp ON, p=0.06). On the right of each scatter plot, kernel density estimation of the initial bar position distribution. Colored shade around the regression line represents SEM.

*Figure 5 continued on next page*

*Figure 5 continued*

The online version of this article includes the following figure supplement(s) for figure 5:

**Figure supplement 1.** Raw unwrapped angular distance traveled by the flies during bar tracking and optogenetic stimulation.

**Figure supplement 2.** Frequency of bar tracking saccades per bin of LED OFF or ON (five progressive bins of 5 s each).

**Figure supplement 3.** Saccade dynamics during bar tracking and optogenetic stimulation.

the three genotypes, the optogenetic activation of T3 neurons was the only perturbation to produce a significant drop in performance (*Figure 5E*).

We next assessed how frequently saccades were triggered under CsChrimson perturbation. T3>CsChrimson flies showed on average an increase frequency of saccades during OFF epochs relative to ON epochs, whereas the frequency of saccades was similar across ON/OFF phases for T4/T5 activated flies and Empty>CsChrimson controls (*Figure 5F*). These results indicate that, in agreement with the PI results (*Figure 5E and F*), population depolarization of T3 neurons perturbed the visual pathway supporting bar-evoked saccades. Yet, when the optogenetic stimulus was switched OFF, recovery from sustained non-selective depolarization strongly revived saccadic bar tracking in flies expressing channelrhodopsin in T3 (*Figure 5F*). To understand whether these flies were restoring the baseline frequency of saccades after the light turned OFF or showing some sort of rebound effect after ON excitation, we segregated the ON-start and OFF-start epochs (*Figure 5—figure supplement 2*). T3>CsChrimson flies starting with an OFF epoch showed a slightly higher saccade rate than controls, suggesting that these flies had a higher baseline saccade rate, rather than any CsChrimson activation rebound effect (*Figure 5—figure supplement 2*). This elevated baseline saccade could reasonably be due to tonic subthreshold depolarization of T3 by the LED display, which could amplify intrinsic synaptic inputs to T3. Irrespective of the small elevated baseline saccade rate, in T3>CsChrimson flies the saccade rate dropped during every ON epoch, and increased during every OFF epoch in a push–pull manner, regardless of the random initial condition. By contrast, no such changes were evident in control or T4/T5>CsChrimson flies (*Figure 5—figure supplement 2*).

We next extracted the bouts of bar tracking behavior. Light-gated depolarization of T3 significantly shortened the duration and number of saccades per bout (*Figure 5G*). A similar trend was shown by T4/T5>CsChrimson notwithstanding the normal tracking performance during ON epochs (*Figure 5G*). This apparent confound is explained by the increased inter-saccadic tracking gain upon optogenetic stimulation of T4/T5 (*Figure 5H*). T4/T5>CsChrimson flies showed stronger smooth responses to bar motion (thereby destabilizing large-field optomotor gaze control, *Figure 5—figure supplement 1*). This destabilization phenomenon is also evident in the positive correlation between PI and bar position relative to the fly's heading at the beginning of each stimulation period (*Figure 5I*).

We next tested whether saccade dynamics (rather than saccade execution) were affected by optogenetic stimulation of T3. In controls and T3 activated flies, saccade angular velocity was similar for OFF and ON optogenetic stimulation epochs, whereas it increased in T4/T5 activated flies during ON epochs compared to OFF epochs (*Figure 5—figure supplement 3A*). Saccade duration was affected neither by CsChrimson expression nor optical condition in any genotype (*Figure 5—figure supplement 3B*). Saccade amplitude did not strongly differ between OFF and ON epochs in any genotype, even though it was tendentially higher during ON periods, suggesting at least some artifact of the red light (*Klapoetke et al., 2014*; *Wu et al., 2016*). These results indicate that normal T3 function is required for triggering saccades to track moving bars (*Figure 4*, *Figure 5B–F*), but the control of their dynamics is likely modulated by both T3 and T4/T5 neurons as well as by some other unidentified neurons (*Figure 4—figure supplement 2*, *Figure 5—figure supplement 3*). The relatively weak effect on saccade dynamics might be spurious stimulus artifacts, considering the fact that saccade speed, amplitude, and duration are pretty stereotyped in either magnetically tethered and freely flying flies (*Fry et al., 2003*; *Mongeau and Frye, 2017*; *Muijres et al., 2015*), and that, once triggered, saccades are not influenced by visual feedback (*Bender and Dickinson, 2006a*).

Flies execute distinct classes of saccades (i.e., different polarity, amplitude, and duration) for bar tracking, small object aversion, and large-field perturbations (*Mongeau et al., 2019*; *Mongeau and Frye, 2017*). To test whether T3 signals are used principally or exclusively to trigger bar tracking saccades, we tested flies with a rotating large-field panorama without a bar object (*Figure 6A*). Our prediction was that if T3 functions selectively for bar tracking saccades, then perturbing them would

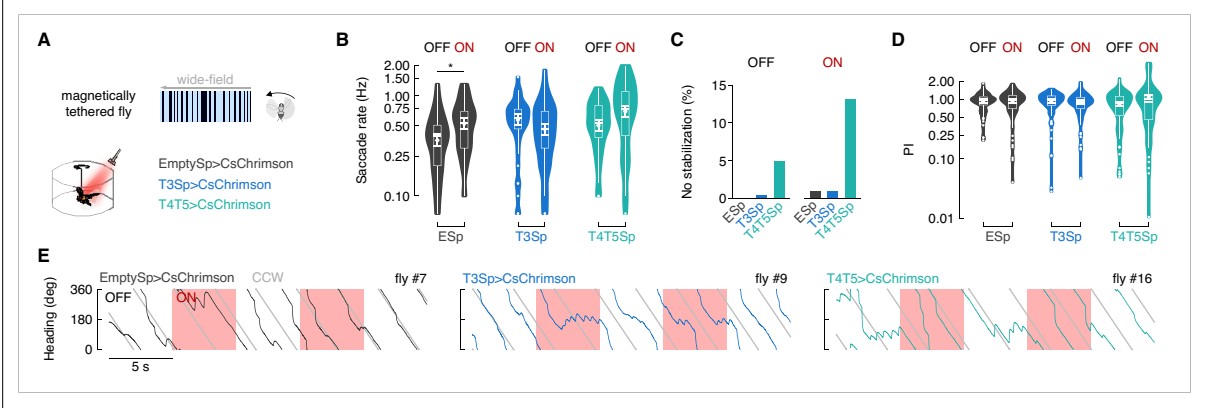

**Figure 6.** Depolarization of T3 does not affect the stabilization of a rotating wide-field panorama. (**A**) Top: representation of the wide-field pattern consisting of bright and dark random stripes as seen by the fly. Bottom: cartoon of the magnetic-tether setup implemented for optogenetic stimulation and genotypes tested (n = 20 EmptySp>CsChrimson, n = 22 T3Sp>CsChrimson, n = 21 T4/T5>CsChrimson). (**B**) Violin-box plots of the frequency of optomotor saccades during periods of LED OFF and ON (y-axis is on a log scale). Big white dot represents the mean, thin horizontal bars indicate SEM, and thick horizontal bar indicates the median. Small white dots on the violin tails represent outliers. A generalized linear mixed model was used to fit the data (residuals were fitted with a gamma distribution and a log link function) and pairwise post-hoc comparisons using *t*-tests adjusted with Bonferroni method were used to compare the predictions (EmptySp OFF vs. ON, p=0.02, Cohen's *d* = –0.73; T3Sp OFF vs. ON, p=0.94, Cohen's *d* = 0.41; T4/T5Sp OFF vs. ON, p=0.09, Cohen's *d* = –0.62). (**C**) Percentage of trials in which flies did not stabilize the rotating wide-field pattern during periods of LED OFF and ON. A performance index (PI) was computed as in *Figure 4D*. Bar plot represents the percentage of trials per genotype where the PI was equal of less than 0. (**D**) Violin-box plots of the PI in trials with positive value (EmptySp OFF vs. ON, p=1, Cohen's *d* = –0.01; T3Sp OFF vs. ON, p=1, Cohen's *d* = 0.12; T4/T5Sp OFF vs. ON, p=0.36, Cohen's *d* = –0.65). Null model was the best model tested. Graph features and statistical approach are as in *Figure 5E*. (**E**) Wrapped heading traces from three representative flies (black: EmptySp>CsChrimson; blue: T3Sp>CsChrimson; green: T4/T5Sp>CsChrimson) responding to a counterclockwise (CCW) rotating wide-field panorama during optogenetic stimulation. Red shaded regions represent periods of LED ON.

The online version of this article includes the following figure supplement(s) for figure 6:

**Figure supplement 1.** Raw unwrapped angular distance traveled by the flies during wide-field stabilization and optogenetic stimulation.

**Figure supplement 2.** Saccade dynamics during wide-field stabilization and optogenetic stimulation.

**Figure supplement 3.** Saccade rate and performance index in bar tracking and wide-field stabilization in flies expressing CsChrimson.

have no influence over large-field saccades. In support of this prediction, the number of large-field optomotor saccades executed by T3>CsChrimson flies did not change significantly between OFF and ON epochs (*Figure 6B*). However, small increases during ON epochs were shown by Empty->CsChrimson and T4/T5>CsChrimson flies, suggesting to us that the red light was likely triggering startle responses (*Figure 6B*). We computed PIs and counted the number of trials in which flies failed to stabilize the large-field rotating panorama (i.e., PI equal or less than zero). As expected, T4/T5>CsChrimson flies showed a higher percentage of no stabilization trials compared to Empty->CsChrimson and T3>CsChrimson flies (*Figure 6C*). Considering only positive PI, controls and T3 activated flies showed no effect in the performance between OFF and ON epochs, while T4/T5 activated flies were more inclined to increase the PI during ON than OFF epochs (*Figure 6D*). This means that the perturbation of T4/T5 neurons could alter the normal performance either by driving an abrupt change in the steering direction or by dramatically increasing the smooth optomotor response (*Figure 6E*, *Figure 6—figure supplement 1*).

The angular velocity of large-field evoked saccades was modestly elevated in T4/T5>CsChrimson flies (*Figure 6—figure supplement 2A*), which was accompanied by reduced saccade duration during ON epochs compared to OFF epochs (*Figure 6—figure supplement 2B*). Saccade amplitude was unaffected for any genotype or optical activation condition (*Figure 6—figure supplement 2C*).

Taken together, these results demonstrate, as expected, that T4/T5>CsChrimson perturbs gaze stabilization in response to both a bar and large-field displacements, whereas the effects of T3>CsChrimson are manifest only in the execution of bar tracking saccades. Interestingly, flies expressing CsChrimson in T3 and T4/T5 but not exposed to the red light for optogenetic activation showed a higher saccade rate for bar tracking than controls, which is the opposite behavior shown by the same driver lines expressing Kir2.1 (*Figure 6—figure supplement 3*). Apparently, the visual

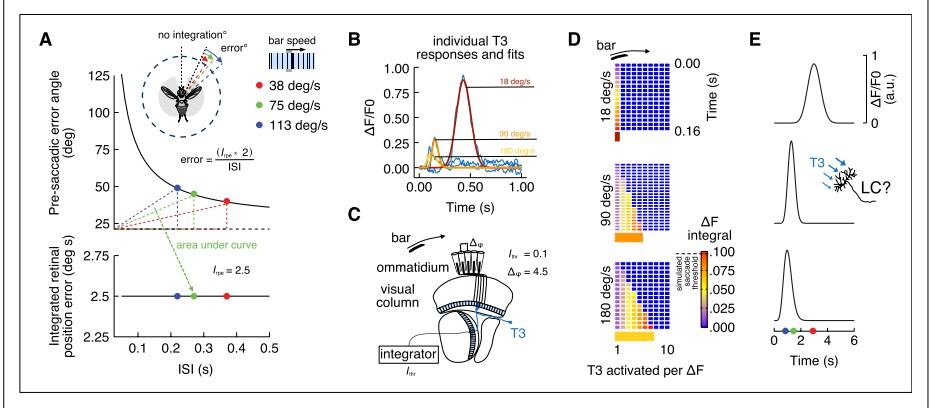

**Figure 7.** Integrate-and-fire model physiologically inspired by T3 calcium dynamics. (**A**) Top: pre-saccadic error angles modeled by using the inverse function of the integrated retinal position error (***Mongeau and Frye, 2017***). Knowing the inter-saccadic intervals (ISI) from previous behavioral experiments as a function of the bar speed and the value of the integrated retinal position error (*Irpr*), we can compute the pre-saccadic error angle as a function of ISI. Modeled error angles are very close to the average errors from behavioral experiments (ISI = 0.22 s, error = 49°; ISI = 0.27 s, error = 45°; ISI = 0.37 s, error = 40°). Dashed black line represents the offset from which the integration of the bar position over time starts (in this model ~26° from visual midline). The area of the triangles defined by the dashed colors lines is constant and represents the *Irpr*. Bottom: integrated retinal position error (Irpr = 2.5° s) derived from previous experiments (***Mongeau and Frye, 2017***) that represents the multiplication of pre-saccadic error angle by ISI. (**B**) Average calcium responses of T3 neurons (blue) to a moving motion-defined bar at three different speeds (data from ***Figure 1I***). These responses were fitted using nonlinear regression analysis (yellow: 180° s⁻¹; orange: 90° s⁻¹; red: 18° s⁻¹). Areas under the fitted curves were computed (*ICa2+*). (**C**) Schematic representation of the fly visual lobe with a 1D organization of T3 neurons that are sequentially activated by a moving bar across the retina. A T3 neuron is present in each column and the interommatidial angle (Δφ) is ~4.5°. We set an arbitrary integrated threshold (Ithr = 0.1) that a downstream partner of T3 would use to trigger a saccade. (**D**) Space-time plots of *Ithr* accumulating over time across cells depending on bar speed (*ICa2+* is based on the physiology, Δφ is known and *Ithr* is arbitrary). Histogram at the bottom indicates the number of T3 cells that in a 1D space would be required to trigger a saccade according to the model. (**E**) Simulated responses by speed of a hypothetical integrator downstream of T3 neurons. LC17 cells might play this role.

The online version of this article includes the following figure supplement(s) for figure 7:

**Figure supplement 1.** Control model for triggering saccades.

display was sufficient to evoke depolarization of T3 and T4/T5, causing enhanced responses to visual cues and combining to improve saccadic bar tracking. We reckon that in T4/T5 this effect facilitated gaze stabilization to enable efficient integration of bar error, whereas in T3 the integration itself was enhanced.

## T3 calcium dynamics support an integrate-and-fire model for triggering saccades

Prior work has shown that bar tracking saccades are triggered neither by absolute retinal position nor by bar velocity, but rather by a threshold in the spatial integral of bar position over time (***Mongeau and Frye, 2017***), or integrated retinal position error ($I_{rpe}$). Therefore, for bar tracking saccades, the fixed $I_{rpe}$ is the product of time (inter-saccadic interval [ISI]) and angular distance traveled by the bar with respect to the fly's heading (pre-saccadic error angle). Accordingly, ISI and pre-saccadic error angle terms are inversely proportional, and $I_{rpe}$ is invariant across bar speed (***Mongeau and Frye, 2017***). When the bar moves fast, the ISI is short and the pre-saccadic error angle is large. When the bar moves slowly, the ISI is extended and the pre-saccadic error angle is reduced (***Figure 7A***).

We tested whether T3's bar-driven calcium responses support a spatiotemporal integrate-and-fire model of bar tracking behavior. Our hypothesis is that calcium signals generated by retinotopic columnar T3 neurons are temporally integrated to reach firing threshold. The number of retinal facets, corresponding neural columns, and T3 neurons that are stimulated by bar motion is proportional to bar speed and inversely related to the visual dwell time and stimulation intensity of T3. During the

integration time, if the bar moves quickly away from the reference position, then many retinal facets would be stimulated with little dwell time, whereas if the bar moves slowly, fewer facets would be stimulated but with larger stimulus dwell time on each. For a downstream integrating neuron, short dwell time corresponds to a low-amplitude signal from any given columnar input, requiring many such signals to charge the integrator to threshold. If the bar moves slowly, larger amplitude signals mean fewer are needed to reach firing threshold. This simple scheme would require neural responses that are inversely proportional to bar speed, which is what we observed in T3 (*Figure 1I*, *Figure 7B*): low-amplitude calcium responses to fast-moving bars, and large responses to slow-moving bars. We fitted nonlinear regressions to the GCaMP responses to simulated bar speeds that bracket behaviorally relevant parameters, and we calculate the requisite integrals ($I_{Ca2+}$) (*Figure 7B*). Then, we used these fits in a model 1D cell array corresponding to the fly's horizontal visual midline (*Figure 7C*). An arbitrary threshold ($I_{thr}$ = 0.1) represents the integrated retinal position error and, since there is one T3 cell per medulla column (*Takemura et al., 2015*), we set the interommatidial angle ($\Delta_\varphi$) to 4.5°. By dividing $I_{thr}$ by $I_{Ca2+}$ then multiplying by $\Delta_\varphi$, we computed the number of T3, their individual activation level required for each bar speed to reach an integrated firing threshold (*Figure 7D*). Presumably, activity of the T3 cell array would be summed by a downstream integrator to the firing threshold for triggering a saccade (*Figure 7E*). This integrated-and-fired mechanism alone, however, does not explain the reduced ISI with increasing bar speed observed in the behavioral data since T3 inputs would be integrated over a quite constant time window (*Figure 7E*). Despite that, a simulated fly's angular position based on this physiologically inspired control system with stereotyped saccade amplitudes and a frontal narrow region of space within which no integration takes place can mimic real fly behavior recapitulating a constant $I_{rpe}$ (*Figure 7—figure supplement 1*). This simple model provides a parsimonious explanation for how T3 neurons could provide behaviorally relevant signals to trigger bar tracking saccades. Moreover, the following predictions can be drawn: (i) a region of the visual field (related to the binocular vision) where the saccades are less likely to be triggered; (ii) an outside region where saccades have the same probability to be performed; (iii) the ISI in this latter region is speed-independent. Finally, cutting-edge all-optical physiology approaches combining calcium imaging and optogenetics in behaving flies could be used to test the predictions about the number of T3 cells necessary to charge to threshold the downstream integrator and trigger a saccade.

## Discussion

With this work, we show that (i) columnar T3 neurons are small-field (local) detectors of the features contained within stimuli that flies readily track during flight (*Figures 1 and 2*), (ii) the integration of these local signals could support the integrated error computations that flies make to track bars (*Figure 7*), which (iii) can explain why manipulation of T3 function compromises bar tracking saccades (*Figures 4 and 5*).

Like virtually every animal with image forming eyes, including humans, flies use smooth optomotor movements to stabilize gaze and to maintain visual course control during locomotion (*Land, 1992*). It is widely accepted that optomotor stabilization reflexes in flies are elicited by patterns of large-field optic flow that is detected by spatially integrating the signals from two identified classes of directionally selective motion detecting columnar neurons, T4 and T5, that have narrow RFs (small-field) that retinotopically sample the entire visual field (*Mauss and Borst, 2020*; *Yang and Clandinin, 2018*). T4/T5 neurons innervate the lobula plate, where their synaptic signals are integrated by large-field neurons such as the horizontal system class (*Shinomiya et al., 2022*) of lobula plate tangential cells (LTPCs). LPTCs have complex directional RFs that act as spatial filters for the patterns of optic flow generated by specific flight maneuvers, and which in turn coordinate smooth optomotor responses (*Busch et al., 2018*; *Krapp and Hengstenberg, 1996*).

In parallel with smooth continuous optomotor movements, flies, like humans, also execute saccades to shift gaze, both during walking (*Geurten et al., 2014*) and in flight to track or avoid salient moving objects (*Mongeau et al., 2019*; *Mongeau and Frye, 2017*). Models of motion-dependent frontal bar fixation do not account for saccadic object pursuit (*Fenk et al., 2014*; *Reichardt and Poggio, 1976*), which instead are believed to be coordinated by a pathway operating in parallel to T4/T5 directional motion detectors (*Aptekar et al., 2012*; *Bahl et al., 2013*). Here, we provide evidence for a component of this parallel pathway; columnar T3 neurons arise from the same optic lobe neuropil as T4/T5, but rather than innervating the lobula plate center for directional motion vision, T3 terminate in the

outer lobula, a center of visual feature detection (*Keleş and Frye, 2017a*). Directional selectivity characteristic of T4/T5 neurons is well suited to control smooth, directional optomotor responses, whereas T3 are better suited to detect the features of visual objects that flies chase or avoid with either syn-directional and counter-directional saccades (*Mongeau et al., 2019*). For example, T4 and T5 are half-wave rectified for contrast increments and decrements, respectively, providing separate ON and OFF motion detection channels, and are tuned to the temporal frequency of a moving pattern (*Creamer et al., 2018*; *Maisak et al., 2013*). By contrast, T3 are distinguished from T4/T5 (and also from neighboring T2a neurons) by full wave rectification (*Keleş et al., 2020*) (i.e., ON-OFF selectivity), omnidirectionality, and sensitivity to object speed largely independently of temporal frequency (*Figure 1*, *Figure 1—figure supplement 1*). Speed tuning could ensure that object processing for orientation saccades is not confounded by spatial structure. T3 is highly sensitive to the high-frequency (finely textured) stimuli that robustly drive bar tracking behavior in flight (*Figure 1*). By contrast, the same stimuli drive T4/T5 to a significantly lesser extent (*Keleş et al., 2018*).

We demonstrate that T3 cells can respond to large-field gratings as well, which is contrary to prior work (*Keleş et al., 2020*; *Tanaka and Clark, 2020*). However, methodological differences include prior work using low-contrast gratings, whereas our stimuli match the behavior experiments using maximum contrast broadband stimuli. Moreover, prior visual stimuli subtended bilaterally over 200° in azimuth, whereas here we provide stimuli unilaterally over a window less than 100° in azimuth. Finally, *Keleş et al., 2020* used a standard Gal4 driver, whereas here we use a split-Gal4 that is highly specific for T3.

Our results demonstrate that T3 cells are not necessarily 'selective' for small objects since they respond to elongated bars and large-field gratings at least when projected only within the ipsilateral visual hemisphere. T3 is, however, more sensitive to small objects: vertical bars yielded a mean response peak ~1 ΔF/F, whereas a small square object elicited a peak of ~4 ΔF/F (*Keleş et al., 2020*). This amplitude differential likely indicates surround inhibition, but does not preclude a downstream integrating neuron from pooling columnar inputs to assemble a spatial RF selective for either an elongated bar or a small object. Of note, individual T4/T5 neurons show roughly double the response amplitude to a small object than a long vertical bar (*Keleş et al., 2020*), which is consistent with other reports. Similarly to T4/T5, which are the first local direction selective detectors of the visual pathway, T3 neurons could represent the first local direction agnostic speed detectors with properties that could go beyond the flicker detection (*Keleş et al., 2020*; *Tanaka and Clark, 2020*). We conclude that T3 neurons do not exclusively work as small object detectors but that they function to provide motion features to a broader group of downstream partners.

T3 form cholinergic chemical synapses (*Konstantinides et al., 2018*) with LC projection neurons LC11 and LC17 (*Tanaka and Clark, 2022*). LC11 is a small object movement detector that plays no known role in flight control, but rather seems to coordinate conspecific social interactions (*Ferreira and Moita, 2020*; *Keleş and Frye, 2017b*). A behavioral screen of freely walking flies showed that optogenetic activation of LC17 induced turning responses (*Wu et al., 2016*), and LC17 has been shown to respond to looming stimuli (*Klapoetke et al., 2022*), but any potential role in saccadic object tracking has not been investigated. A recent volumetric imaging of the VPNs at population level in walking flies has shown that LC17 cells respond strongly to vertical moving bars (*Turner et al., 2022*). Future experiments will try to pinpoint the downstream integrator of T3 neurons.

To our knowledge, this is the first identification of a visual neuron type that specifically serves saccadic bar tracking behavior. However, other types of columnar neurons might contribute to this and other similar visual behaviors (e.g., T2a and T2 neurons). Our evidence supports the view that T4/T5 play a fundamental role in maintaining stable gaze, which seems to be a prerequisite for robust saccadic tracking behavior. Moreover, T4/T5 might coordinate distinct components of the dynamics of this behavior, possibly via identified projection neurons that interconnect the lobula plate and deeper layers of the lobula (*Shinomiya et al., 2022*). A control theoretic model for saccadic bar tracking requires two key computations: (i) a discrimination between exafferent and reafference motion, which depends upon direction (*Heisenberg and Wolf, 1988*), and (ii) an integrate-and-fire threshold operation to trigger a saccade, which depends upon the accumulation of retinal error (*Mongeau and Frye, 2017*).

**Table 1.** Origin of reagents used in this study.

| Reagent type | Source | Identifier |
|---|---|---|
| *D. melanogaster*: Empty-SplitGal4 [p65ADZp.Uw(attP40); ZpGDBD.Uw(attP2)] | Bloomington *Drosophila* Stock Center | BDSC #79603 |
| *D. melanogaster*: T3-SplitGal4 [VT002055-p65ADZp(attP40); R65B04- ZpGDBD(attP2)] | *Keleş et al., 2020* | N/A |
| *D. melanogaster*: T2a-SplitGal4 [VT012791-p65ADZp(attP40); R47E02- ZpGDBD(attP2)] | *Keleş et al., 2020* | N/A |
| *D. melanogaster*: T4/T5-SplitGal4 [R59E08-p65.AD(attP40); R42F06-Gal4.DBD(attP2)] | G.Rubin | N/A |
| *D. melanogaster*: 20XUAS-IVS-GCaMP6f(VK00005) | Bloomington *Drosophila* Stock Center | BDSC #52869 |
| *D. melanogaster*: 20XUAS-IVS-jGCaMP7f(VK00005) | Bloomington *Drosophila* Stock Center | BDSC #79031 |
| *D. melanogaster*: 10XUAS-IVS-eGFP-Kir2.1(attP2) | *von Reyn et al., 2017* | N/A |
| *D. melanogaster*: 20xUAS-Chrimson::tdTomato(VK00005) | D.Anderson | N/A |
| Antibody: anti-GFP (chicken polyclonal) | Abcam | Cat#: ab13970; RRID:AB_300798 |
| Antibody: anti-Brp (mouse monoclonal) | Developmental Studies Hybridoma Bank | Cat#: nc82; RRID:AB_2314866 |
| Antibody: anti-chicken, Alexa Fluor 488 (goat polyclonal) | Thermo Fisher Scientific | Cat#: A11039; RRID:AB_2534062 |
| Antibody: anti-mouse, Alexa Fluor 647 (goat polyclonal) | Thermo Fisher Scientific | Cat#: A21236; RRID:AB_2535804 |

Our results suggest that directional motion detectors T4 and T5 coordinate a pathway involved in keeping the fly still if it is moving relative to the background (*Figures 4 and 5*) and in keeping the background still if it is moving relative to the fly (*Figure 4—figure supplement 3*). Our results also suggest that a parallel pathway based on T3 neurons participates in triggering saccades because their synaptic perturbation suppresses both orientation steering effort by the wings (*Figure 3*) and object pursuit body saccades (*Figures 4 and 5*). Finally, a model based on the integrated output of T3 neurons captures the spatiotemporal threshold dependence of bar tracking behavior (*Figure 7*). Our working model is no doubt an incomplete accounting of the full underlying control interactions. Nevertheless, we have for the first time provided a strong conceptual framework, and identified its key elements, for the parallel neural control of smooth optomotor stabilization and saccadic bar tracking control systems within a key model system.

## Methods

### Fly strains

For all experiments we used 3–5-day-old female *Drosophila melanogaster* reared on standard corn-meal molasses at 25°C, 30–50% humidity entrained to 12 hr light/12 hr dark cycle. All behavioral experiments involving silencing of targeted neurons were performed with flies carrying at least one wild-type *white* allele. We were not blind to genotype, and the tested flies were randomly chosen from the vials. Fly lines and their origins are listed in *Tables 1 and 2*.

### Imaging setup

Two-photon calcium imaging was performed on a modified upright microscope (Axio Examiner, Zeiss), exciting the specimens with a Ti:Sapphire laser (Chameleon Vision, Coherent) tuned to 920 nm (power at the back aperture ~25 mW). We imaged with a ×20 water-immersion objective (W Plan-Apochromat ×20/1.0 DIC, Zeiss). Data acquisition was controlled by Slidebook (version 6, 3i). Single-plane images

**Table 2.** Genotypes and experimental parameters used in each figure.

| Description | Genotype | Experiment | Figure # | Flies # | Repetition # |
|---|---|---|---|---|---|
| T3Sp>GCaMP6f | w⁻/+; VT002055-AD/+; R65B04-DBD/20xUAS-GCaMP6f | ON and OFF moving bars (9° width) in two directions (front-to-back and back-to-front) at three different speeds (18° s⁻¹, 90° s⁻¹, 180° s⁻¹) | *Figure 1D and J; Figure 1—figure supplement 1* | 11 | 3 |
| T3Sp>GCaMP6f | w⁻/+; VT002055-AD/+; R65B04-DBD/20xUAS-GCaMP6f | Single-pixel OFF bar (2.25° width) moving in four different directions (upward, downward, leftward, and rightward) at 18° s⁻¹ | *Figure 1G and H* | 5 | 1 |
| T3Sp>GCaMP6f | w⁻/+; VT002055-AD/+; R65B04-DBD/20xUAS-GCaMP6f | Motion-defined bars (9° width) moving in two directions (front-to-back and back-to-front) at three different speeds (18° s⁻¹, 90° s⁻¹, 180° s⁻¹) | *Figure 1I* | 11 | 3 |
| T2aSp>GCaMP6f | w⁻/+; VT012791-AD/+; R47E02-DBD/20xUAS-GCaMP6f | Single-pixel OFF bar (2.25° width) moving in four different directions (upward, downward, leftward, and rightward) at 18° s⁻¹ | *Figure 1—figure supplement 2D and E* | 4 | 1 |
| T2aSp>GCaMP6f | w⁻/+; VT012791-AD/+; R47E02-DBD/20xUAS-GCaMP6f | ON, OFF, and motion-defined moving bars (9° width) in two directions (front-to-back and back-to-front) at three different speeds (18° s⁻¹, 90° s⁻¹, 180° s⁻¹) | *Figure 1—figure supplement 2F and G* | 9 | 3 |
| T2aSp>GCaMP6f | w⁻/+; VT012791-AD/+; R47E02-DBD/20xUAS-GCaMP6f | Gratings patterns of $\lambda$ = 18° and $\lambda$ = 36° moving at two different temporal frequencies (1 Hz and 2 Hz) in two directions (front-to-back and back-to-front) | *Figure 1—figure supplement 2H* | 9 | 3 |
| T3Sp>GCaMP6f | w⁻/+; VT002055-AD/+; R65B04-DBD/20xUAS-GCaMP6f | Gratings patterns of $\lambda$ = 18° and $\lambda$ =36° moving at three different temporal frequencies (1 Hz, 2 Hz, and 5 Hz) in two directions (front-to-back and back-to-front) | *Figure 2A–C* | 11 | 3 |
| T3Sp>jGCaMP7f | w⁻/+; VT002055-AD/+; R65B04-DBD/20xUAS-jGCaMP7f | Gratings patterns of six different spatial frequencies ($\lambda$ = 9°, 18°, 27°, 36°, 45°, 54°) moving at six different temporal frequencies (Hz = 1, 2, 3, 5, 7, 10) in two directions (front-to-back and back-to-front) | *Figure 2F–J; Figure 2—figure supplement 1* | 10 | 1 |
| EmptySp>Kir2.1 | w⁻/+; empty-AD/+; empty-DBD/10xUAS-Kir2.1 | Motion-defined and dark bars (18° width) moving at 90° s⁻¹ CW and CCW and starting from behind the fly | *Figure 3B and C; Figure 3—figure supplement 2B and D* | 44 | 3 |
| T3Sp>Kir2.1 | w⁻/+; VT002055-AD/+; R65B04-DBD/10xUAS-Kir2.1 | Motion-defined and dark bars (18° width) moving at 90° s⁻¹ CW and CCW and starting from behind the fly | *Figure 3B and C; Figure 3—figure supplement 2B and D* | 26 | 3 |
| T4/T5Sp>Kir2.1 | w⁻/+; R59E08-AD/+; R42F06-DBD/10xUAS-Kir2.1 | Motion-defined and dark bars (18° width) moving at 90° s⁻¹ CW and CCW and starting from behind the fly | *Figure 3B and C; Figure 3—figure supplement 2B and D* | 22 | 3 |

*Table 2 continued on next page*

*Table 2 continued*

| Description | Genotype | Experiment | Figure # | Flies # | Repetition # |
|---|---|---|---|---|---|
| EmptySp>Kir2.1 | w⁻/+; empty-AD/+; empty-DBD/10xUAS-Kir2.1 | Dark and motion-defined bars in closed-loop for 60 s | *Figure 3—figure supplement 3* | 28 | 1 |
| T3Sp>Kir2.1 | w⁻/+; VT002055-AD/+; R65B04-DBD/10xUAS-Kir2.1 | Dark and motion-defined bars in closed-loop for 60 s | *Figure 3—figure supplement 3* | 26 | 1 |
| EmptySp>Kir2.1 | w⁻/+; empty-AD/+; empty-DBD/10xUAS-Kir2.1 | Motion-defined bar (18° width) and large-field panorama moving at 112.5° s⁻¹ CW and CCW (bar started at random locations) | *Figure 4B–J; Figure 4—figure supplement 2; Figure 4—figure supplement 3* | 23 | 1 |
| T3Sp>Kir2.1 | w⁻/+; VT002055-AD/+; R65B04-DBD/10xUAS-Kir2.1 | Motion-defined bar (18° width) and large-field panorama moving at 112.5° s⁻¹ CW and CCW (bar started at random locations) | *Figure 4B–J; Figure 4—figure supplement 2; Figure 4—figure supplement 3* | 22 | 1 |
| T4/T5Sp>Kir2.1 | w⁻/+; R59E08-AD/+; R42F06-DBD/10xUAS-Kir2.1 | Motion-defined bar (18° width) and large-field panorama moving at 112.5° s⁻¹ CW and CCW (bar started at random locations) | *Figure 4B–J; Figure 4—figure supplement 2; Figure 4—figure supplement 3* | 22 | 1 |
| EmptySp>CsChrimson | w⁻/+; empty-AD/+; empty-DBD/20xUAS-Chrimson | Motion-defined bar (18° width) and large-field panorama moving at 112.5° s⁻¹ CW and CCW (bar started at random locations) during optogenetic stimulation | *Figure 5C–I; Figure 5—figure supplement 2; Figure 5—figure supplement 3; Figure 6B–D; Figure 6—figure supplement 2* | 20 | 1 |
| T3Sp>CsChrimson | w⁻/+; VT002055-AD/+; R65B04-DBD/20xUAS-Chrimson | Motion-defined bar (18° width) and large-field panorama moving at 112.5° s⁻¹ CW and CCW (bar started at random locations) during optogenetic stimulation | *Figure 5C–I; Figure 5—figure supplement 2; Figure 5—figure supplement 3; Figure 6B–D; Figure 6—figure supplement 2* | 22 | 1 |
| T4/T5Sp>CsChrimson | w⁻/+; R59E08-AD/+; R42F06-DBD/20xUAS-Chrimson | Motion-defined bar (18° width) and large-field panorama moving at 112.5° s⁻¹ CW and CCW (bar started at random locations) during optogenetic stimulation | *Figure 5C–I; Figure 5—figure supplement 2; Figure 5—figure supplement 3; Figure 6B–D; Figure 6—figure supplement 2* | 21 | 1 |
| EmptySp>CsChrimson | w⁻/+; empty-AD/+; empty-DBD/20xUAS-Chrimson | Motion-defined bar (18° width) and large-field panorama moving at 112.5° s⁻¹ CW and CCW (bar started at random locations) | *Figure 6—figure supplement 3* | 19 | 1 |
| T3Sp>CsChrimson | w⁻/+; VT002055-AD/+; R65B04-DBD/20xUAS-Chrimson | Motion-defined bar (18° width) and large-field panorama moving at 112.5° s⁻¹ CW and CCW (bar started at random locations) | *Figure 6—figure supplement 3* | 18 | 1 |
| T4/T5Sp>CsChrimson | w⁻/+; R59E08-AD/+; R42F06-DBD/20xUAS-Chrimson | Motion-defined bar (18° width) and large-field panorama moving at 112.5° s⁻¹ CW and CCW (bar started at random locations) | *Figure 6—figure supplement 3* | 19 | 1 |

*Table 2 continued on next page*

*Table 2 continued*

| Description | Genotype | Experiment | Figure # | Flies # | Repetition # |
|---|---|---|---|---|---|

CW: clockwise; CCW: counterclockwise.

were taken at ~10 Hz with a spatial resolution of approximately 285 × 142 pixels (100 × 50 μm, pixel size ≅ 0.35 μm, dwell time ≅ 2.5 μs). GCaMP responses were recorded from the presynaptic terminals of T3 neurons. For each preparation, we identified the most caudal presynaptic terminals and then shifted the region of interest (ROI) downward ~30 μm. Images and external stimulations were synchronized a posteriori using frame capture markers (TTL pulses output from Slidebook) and stimulus events (analog outputs from the LED display controller) sampled with a data acquisition device (DAQ) (PXI-6259, NI) at 10 kHz. The DAQ interfaced with MATLAB (R2020a, MathWorks) via rack-mount terminal block (BNC-2090, NI).

## Fly preparation for imaging

Flies were cold anesthetized at ~4°C using a thermoelectric cooling system and mounted on a custom 3D printed fly holder (*Weir and Dickinson, 2015*). Specifically, flies were gently pushed through a hole etched on a stainless-steel shim so that the dorsal thorax protruded from the dorsal aspect of the horizontally mounted shim and the ventral thorax with the abdomen remained below. The head was pitched forward to expose its posterior surface without stretching the neck connective. Flies were secured using UV-curable glue (44600, Dreve Fotoplast Gel) around the posterior–dorsal cuticle of the head capsule and across the dorsal thorax. To reduce movement artifacts during the recordings, we immobilized the legs and the proboscis with low melt point bees-wax (Waxlectric-1, Renfert). We used fine forceps (Dumont #5SF, Fine Science Tools) to remove the posterior cuticle, fat bodies, and post-ocular air-sac obstructing the view of the right optic lobe. We severed muscles 1 and 16 to reduce brain movements (*Demerec, 2008*). The brain was bathed in physiological saline containing (in mM) 103 NaCl, 3 KCl, 1.5 CaCl$_2$, 4 MgCl$_2$, 26 NaHCO$_3$, 1 NaH$_2$PO$_4$, 5 N-Tris (hydroxymethyl) methyl-2-aminoethane-sulfonic acid (TES), 10 trehalose, 10 glucose, 2 sucrose. The saline was adjusted to an osmolarity of 273–275 mOsm and a pH of 7.3–7.4. The brain was continuously perfused with extracellular saline at 1.5 ml/min via a gravity drip system and the bath was maintained at 22°C by an inline solution heater/cooler (SC-20, Warner Instruments) and temperature controller (TC-324, Warner Instruments).

## Visual stimuli for imaging

During two-photon imaging, visual stimuli were presented on an LED display (*Reiser and Dickinson, 2008*) composed of 48 panels arranged in a semi-cylinder (Panels arena, IO Rodeo). The display covered ±108° in azimuth and ±32° in elevation. Each LED subtended a visual angle of ~2.25°. To reduce the light intensity from the LED display, three layers of filter (R59-indigo, Rosco) were placed over the display. The display had, at its maximum intensity, an irradiance of ~0.11 μW m$^{-2}$ (recorded at the fly's position) at the spectral peak of 460 nm (full width at half maximum: 243 nm). Visual patterns were generated and controlled using custom-written MATLAB scripts that communicated to a custom-designed controller via a serial port, which in turn communicated to the panels via a rapid serial interface https://reiserlab.github.io/Modular-LED-Display/G3/ (*Reiser and Dickinson, 2008*). To account for the angle the fly's head had when mounted on the holder, the display was tilted 30° from the horizontal plane. We recorded from the right optic lobe, and the stimulus coordinates are referred to the fly's head position (*Figure 1C*). Therefore, the azimuthal and the elevation position of the stimuli are centered to the fly's visual equator and prime meridian. Unstable recordings or recordings from flies that died during the experiment were no longer considered for the analysis. Visual stimulation was confined to the right half of the visual field, ipsilateral to the recording site. Due to the far peripheral blind spot generated by the imaging stage, visual stimuli were presented within a restricted window of 72° × 72° with the left edge abutting the visual prime meridian. The brightness of the display background, outside the stimulus window, was set to 50% maximum. The stimulus set was presented in random block design, repeated three times. Each visual stimulation lasted 7.5 s and was composed by 0.5 s of uniform background (50% maximum intensity), 0.5 s of static pattern onset

within the stimulation window, variable duration (depending on the stimulus speed) visual motion (maximum 4.5 s) followed by 2 s lingering static pattern. After each visual stimulation, 2 s of rest with the display off (0% of maximum intensity) interspersed the trials to prevent adaptation.

## Receptive field mapping and directional selectivity

As previously described (*Städele et al., 2020*), to characterize the functional RF of T3 and T2a neurons, we multiplied the time series of responses to a 2.25° × 72° (width × height) dark bar (i.e., single pixel stripe) moving horizontally with the time series of responses to a 72° × 2.25° (width × height) dark bar moving vertically. In each preparation, we imaged from the presynaptic terminal of an individual T3 neuron. The single-pixel dark stripe moved within a 9° spaced bin (4 pixels) in all four cardinal directions (upward, downward, leftward, and rightward) at 18° s$^{-1}$, allowing us to extract directional selectivity information. Sweep trials of different directions were randomly presented over a region of the visual field between 0° and 90° in azimuth and between –36° and +36° in elevation. We divided this region in 10 horizontal bins (each one of 9° × 72°) and 8 vertical bins (each one of 90° × 9°). Each sweep (36 in total) was composed of 6 s of rest period with a uniform bright background (50% of maximum intensity) and 0.5 s of stripe motion within a bin. Since we did not observe differences in the directional selectivity of T3 and T2a for horizontal or vertical sweeps, leftward and rightward movement responses were averaged, as well as downward and upward responses. To assess the RF size, these averaged time series for the 10 horizontal bins and the 8 vertical bins were multiplied together, obtaining a matrix of activity (*Figure 1F*). The matrix for each fly was then simplified by taking the values of activity peak and spatially normalized to the bin showing the maximum activity peak (RF center). Finally, the peak values were normalized to the value at the center of the RF (bin #0). We averaged the normalized matrices for each fly to obtain a heatmap of the RF for T3 neurons (*Figure 1G*). The directional selectivity was estimated by considering the activity peaks of the RF center for each sweep separately and plotting them in a polar plot (*Figure 1H*). Flies whose anteroposterior imaging plane moved during the experiment were no longer considered for the analysis.

## Speed tuning

Square-wave gratings were presented within a 72° × 72° window on the right side (ipsilateral) of the LED display (0° offset from visual midline). Combinations of six different spatial wavelengths and six different temporal frequencies established the set of stimuli. Each trial was composed of 2 s of uniform background (50% maximum intensity), 0.5 s of static pattern onset, 2 s of visual motion followed by 2 s lingering static pattern. Per fly, each condition was presented only once and a single ROI around the presynaptic terminal of an individual T3 neuron was drawn. In order to compare different linear models with or without the regression coefficient, we computed the Bayesian information criterion (BIC) for each model (*Schwarz, 1978*). Subsequently, an approximation of the Bayes factor (BF) based on the difference between models' BIC was calculated to obtain the probability of one model over the other (*Raftery, 1995*). The slope of the spatiotemporal plots corresponded to the regression coefficient of a linear model fitted to the data.

## Imaging data analysis

Image stacks were exported from Slidebook (version 6, 3i) in (16-bit) .tiff format and imported into MATLAB for analysis. A user-friendly custom toolbox developed by Ben J. Hardacastle https://github.com/bjhardcastle/SlidebookObj (*Hardcastle, 2021* ; *Hardcastle et al., 2021*) allowed us to correct the images for motion artifacts along the x-y plane using a DFT-based registration algorithm (*Guizar-Sicairos et al., 2008*) and to manually draw a ROI around the presynaptic terminal of an active neuron located approximately in the middle of the lobula. We opted for a manual identification of the ROI to specifically consider morphology and position of the neuron investigated. Through this approach, combined with the consistent z-position of our imaging plane, we were able to identify T3 neurons across preparations at the same location in the neuropile, exhibiting similar spatial RFs. A time series was generated by calculating the mean fluorescence intensity of pixels within the ROI in each frame (F$_t$). These mean values were then normalized to a baseline value as ΔF/F = (F$_t$ – F$_0$)/F$_0$, where F$_0$ was the mean of F$_t$ during the 0.5 s preceding stimulus onset. To compute average time series across preparations with small variations in TTL synchronization, traces were resampled using linear interpolation. This procedure did not cause any detectable change in the original data.

## Rigid-tether setup

In the rigid tether paradigm, flies were cold anesthetized at ~4°C and tethered to tungsten pins (diameter: 0.2 mm) using UV-curable glue (Watch Crystal Glue Clear). The pin was placed on the dorsal thorax so as to get a final pitch angle of the fly (between 35° and 45°) similar to body angle and wing stroke plane during free hovering (*Fry et al., 2003*). Before running the experiment, flies were left to recover upside-down in a custom-designed pin holder, in turn placed into a covered acrylic container, for ~30 min at room temperature (~22°C). Inside the container, we put a small bowl filled with water to maintain humidity and avoid desiccation. To reduce flight energy expenditure, as soon as flies recovered from the anesthesia and started flying, we gently offered them a small square piece of paper (Kimwipes, Kimberly-Clark) ~3 mm side, that they generally clung to with their legs without flying. Flies were then positioned in the center of a cylindrical LED panel display (*Reiser and Dickinson, 2008*) that covered ±165° in azimuth and ±47° in elevation. The display was composed of 96 (horizontal) × 32 (vertical) LEDs (emission peak: 568 nm) each LED subtending 3.75° on the fly's retina. Flies were illuminated from the top with an infrared diode (emission peak: 880 nm), which cast a shadow of the beating wings onto an optical sensor. An associated 'wing-beat analyzer' (JFI Electronics Laboratory, University of Chicago) converted the optical signal into an instantaneous voltage measuring right and left wing beat amplitude (WBA) and frequency (WBF). The difference in the left and right WBA (ΔWBA), which is highly correlated with the fly's steering effort in the yaw axis (*Tammero et al., 2004*), connected to the panel display controller to close a feedback loop with the rotational velocity of the visual display. Noteworthy, the translational velocity of the wings during flight is proportional to the product of the sum between left and right WBA (ΣWBA), which is in turn proportional to the flight force (or thrust), and WBF (*Lehmann and Dickinson, 1997*). The aerodynamic power output produced by the wings is roughly proportional to the cube of the translational velocity (*Duistermars et al., 2007*; *Lehmann and Dickinson, 1997*). In our experiments, flies were exposed to periods of closed-loop where they could control the visual panorama and periods of open-loop where they could not. Signals from the wing-beat analyzer and from the panel display controller, encoding the visual display position, were recorded on a DAQ (Digidata 1440A, Molecular Devices) at 1 kHz. The data acquisition was triggered through a voltage step sent by a second DAQ (USB-1208LS, Measurement Computing) interfaced with MATLAB that in turn controlled the pattern presentations. For silencing experiments, flies expressing Kir2.1 tagged with green fluorescence protein (GFP) were dissected after the behavioral recordings under a fluorescence stereomicroscope (SteREO Discovery.V12, Zeiss) to confirm the expression in the targeted cells.

## Rigid-tether visual stimuli

Each trial was composed of 5 s in closed-loop with a bar and a variable time in open-loop test depending on the speed of the stimuli presented to the fly. The closed-loop periods ensured the fly was engaged in the task, while the open-loop periods were tested for responses to the visual stimuli. A full set of stimuli was randomized and repeated three times, with a total duration of ~5 min. Flies that stopped flying during the experiment or that failed to frontally fixate a dark bar on a uniform bright background during a pre-experiment assessment period were not included in the analysis. All stimuli used in these experiments had patterns of bright (100% of maximum intensity) and dark pixels (0% of maximum intensity). The luminance-defined bar represented a 30° × 94° (width × height) dark bar moving on a uniform bright background. The motion-defined bar represents a 30° × 94° bar of random bright and dark vertical stripes moving over an analog static background of randomly distributed bright and dark stripes. This pattern was generated using a custom-written MATLAB code as previously described (*Keleş et al., 2018*). Briefly, the random-stripes patterns had an equal number of bright and dark pixels and a high-pass filter ensured that no bright or dark contiguous stripes exceeded 22.5° in width (six stripes). A random-stripes pattern was randomly picked up, every time it was needed, from 96 different options to avoid pattern-specific behavioral artifacts. Clockwise (CW) and counterclockwise (CCW) directions were always considered in bar revolving experiments. However, assuming bilateral symmetry, we reflected the time-series responses to CCW stimuli and pooled them with CW responses. At the end of the experiment, flies were exposed to luminance- and motion-defined bars (picked in random order) in closed-loop for 60 s.

## Magnetic-tether setup

In the magnetic tether paradigm, flies were cold anesthetized as for the rigid tether paradigm and glued to stainless steel pins (diameter: 0.1 mm, Fine Science Tools) as previously described (*Bender and Dickinson, 2006b*; *Duistermars and Frye, 2008*; *Mongeau and Frye, 2017*). Briefly, pins were trimmed to be 1 cm length and placed on the dorsal thorax in order to get a final fly's pitch angle of approximately 30°. Flies were then left to recover for ~30 min upside-down sticking out of a polystyrene block. To reduce flies' fatigue, as done for the rigid tether paradigm, flies were provided with a small piece of paper. The visual stimulation was performed using a cylindrical LED panel display covering 360° in azimuth and 56° in elevation (array of 96 × 16 LEDs, emission peak: 470 nm). At the display horizontal midline, each LED subtended an angle of 3.75° on the fly's retina. The fly was suspended between two magnets that maintained the animal in place and free to rotate about its vertical axis (i.e., yaw axis). The pin was attracted at its top end toward the upper magnetic pivot and its moment of inertia encompassed less than 1% of the fly's moment of inertia (*Bender and Dickinson, 2006b*; *Fry et al., 2003*). The fly was illuminated from below with an array of eight infrared diodes (emission peak: 940 nm) and recorded from the bottom with an infrared-sensitive camera (Blackfly S USB3, Teledyne FLIR) fitted with a zoom lens (InfiniStix 0.5x/0.024, Infinity Photo-Optical) at 200 frames s⁻¹. The lens also held a long pass filter to block the light emitted by the display (NIR, Edmund Optics). At the beginning of each trial, after 10 s of acclimatation, the fly was presented for 20 s with a rotating large-field panorama, which elicited strong optomotor responses, in each direction (CW and CCW). Flies whose behavior was characterized by excessive wobble indicating poor tethering were discarded.

## Magnetic-tether visual stimuli

Visual stimuli consisted in a motion-defined bar (30° × 56°) and a large-field panorama (360° × 56°) of random bright and dark vertical stripes rotating horizontally around the fly in CW and CCW directions at 112.5° s⁻¹. We decided to use these stimuli because previous experiments conducted in our laboratory showed that motion-defined bars elicit a higher number of body-saccades compared to solid luminance-defined bars (~50% more) (*Mongeau and Frye, 2017*). For similar reasons, the speed selected elicits robust bar tracking behavior (*Mongeau and Frye, 2017*). Both stimuli were generated, as mentioned above, using custom-written MATLAB code and randomly chosen during each trial among 96 different patterns. Each trial involved 25 s of a rotating stimulus at constant speed and 5 s of resting period in which the stimulus stopped moving. For the bar, the initial position was selected from a pseudo-random sequence. Each fly was tested in four different and randomized trials (2 stimuli × 2 directions) without repetitions to minimize habituation. The full experiment lasted ~3 min. Only flies that either flew continuously or stopped briefly only once were included in the analysis.

## Optogenetic stimulation

Optogenetic experiments were conducted in a setup similar to the one used for magnetic-tether experiments. Likewise, a cylindrical LED panel display (array of 96 × 16 LEDs, emission peak: 470 nm) was used and two layers of neutral filter were placed over the display to reduce the light intensity. A red LED (emission peak: 685 nm, 4 V) was positioned laterally to the post bearing the magnetic pivot within which the fly was suspended. Its beam covered the entire fly and, since the fly freely rotated around the yaw axis, the angle of illumination varied during flight. We used the same visual stimuli, speed, directions, and trial duration used in magnetic-tether experiments. On top of visual rotating stimuli, flies were exposed to contiguous epochs of LED ON and OFF which lasted 5 s each. The starting optogenetic epoch was randomly selected so that flies could start the trial with the LED ON or OFF. This means that during the 25 s of visual stimulation, flies could be optogenetically stimulated for either 10 s or 15 s in total. After every stimulation, flies were left to rest for 5 s facing a static random-stripes pattern. Moreover, we included trials where the LED remained OFF throughout the visual stimulation. The experiment lasted ~6 min (30 s × 3 LED intensities × 2 stimuli × 2 directions). As done for the magnetic-tether experiments, we discarded flies that showed signs of poor tethering preparation (i.e., asymmetric wing beat amplitude). All-trans-retinal (ATR) is required to get a proper CsChrimson protein conformation. In order to boost flies' performance, although flies endogenously produce retinal, we added ATR to the food. The progeny from the crosses between driver lines and

UAS-CsChrimson effector were raised in the darkness to avoid the channel's stimulation. After eclosion, newborn flies were transferred in 0.5 mM ATR food and kept there for 3–5 d until the experiment.

## Immunostaining and confocal microscopy

Female flies were dissected in Schneider's Insect Medium (S0146, Sigma-Aldrich) using fine forceps (Dumont #5SF, Fine Science Tools). Brains and VNCs were fixed with 4% v/v PFA (158127, Sigma-Aldrich) in PBS for 30 min at room temperature. Next, brains and VNCs were washed out of fixative with PBS, solubilized in PBST (0.5% Triton-X100, T9284, Sigma-Aldrich) for 1 hr, and blocked in 7.5% v/v NGS (Normal Goat Serum, 005-000-121, Jackson ImmunoResearch) in PBST for 1–2 hr at room temperature. Brains and VNCs were then incubated in primary antibodies: chicken anti-GFP (1:1000, ab13970, Abcam) and anti-Brp (1:10, nc82, Developmental Studies Hybridoma Bank) for 4 hr at room temperature and 24 hr at 4°C. Samples were subsequently washed out of primary antibodies at least three times with PBST over 2 hr and incubated in secondary antibodies: goat anti-chicken conjugated to Alexa 488 (1:1000, A11039, Thermo Fisher Scientific) and goat anti-mouse conjugated to Alexa 647 (1:200, A21236, Thermo Fisher Scientific) for 4 hr at room temperature and for 48 hr at 4°C. After washing the samples at least three times with PBST over 2 hr, they were finally incubated in VectaShield (H-1000, Vector Laboratories) for 10 min at room temperature, and mounted onto slides for imaging. Brains and VNCs were imaged with a confocal microscope (LSM700 Axio Imager M2, Zeiss) using ZEN software (black edition, Zeiss). Complete series of optical sections were taken at 1 μm intervals with ×10 and ×40 oil immersion objectives. Images were processed using ImageJ (*Schneider et al., 2012*).

## Behavioral analysis

Data collected either in the rigid-tether or in the magno-tether setups were analyzed in RStudio (*RStudio Team, 2021*) using custom R scripts. Axon binary files (.abf) from the DAQ used in rigid tether experiments were imported by using the R package *abf2* (*Caldwell, 2015*) and pre-processed (data filtering) as previously described (*Keleş et al., 2018*). Video recordings from the camera on the magno-tether setup were imported into MATLAB and the flies' heading offline tracked by using custom scripts. Video heading files and DAQ files were then imported into RStudio by using the R package *R.matlab* (*Bengtsson, 2018*). Saccade detection and tracking bouts were analyzed as previously described (*Mongeau and Frye, 2017*). Data plotting was performed by R package *ggplot2* (*Wickham, 2016*).

## Model

In the integrator-and-fire model physiologically-inspired on T3 neurons, the GCaMP responses were fitted by using nonlinear regression analysis (*Bates and Watts, 1988*) with a least-squares method in RStudio (*RStudio Team, 2021*). We used the *nls* function with a predefined model formula inspired to the probability density function of a beta distribution $y = \left( x^{(\alpha - 1)} \times (1 - x)^{(\beta - 1)} \right) \times \gamma$, where $\alpha$ and $\beta$ are shape parameters, while $\gamma$ is a scaling factor. These models minimized the number of parameters, while maintaining wide flexibility and goodness of fit. The behavioral control model was simulated in Simulink (MATLAB).

## Statistics

Linear and generalized mixed effects (GLME) models were used to fit the data in order to consider the random effects represented by individual flies (*DeBruine and Barr, 2021*; *Saravanan et al., 2020*). GLME models avoid averaging that reduces the statistical power and allows for adjusting the estimates for repeated sampling and for sample imbalancing. We fitted the data using the R package *lme4* (*Bates et al., 2015*). Residuals of data normally distributed were fitted with a Gaussian distribution, whereas residuals of strongly skewed data were fitted with a gamma distribution and a log link function. The log transformation is operated on the mean and not on the variable to avoid affecting the variance of the residuals. Pairwise post-hoc comparisons on the fixed effects of the models through *t*-test adjusted with Bonferroni method were performed using the R package *emmeans* (*Lenth, 2021*). With the same R package, we computed the standardized effect size (i.e., Cohen's *d*) on the estimated marginal means and their standard deviation obtained from the models. ANOVA was also computed on the parameters estimated by the models. In violin-box plots, mean and median were

reported as central tendency measures, bottom and top edges of the box represent 25th (Q1) and 75th (Q3) percentiles, and whiskers represent the lowest and highest datum within 1.5 interquartile range (Q3–Q1).

## Acknowledgements

We thank Martha Rimniceanu and Lesly Palacios Castillo for producing confocal images and commenting on the manuscript. We thank Ivan Lopez for collecting preliminary results, Mehmet Keleş for reagents, and Ben Hardcastle for technical advice. This work was supported by a grant from the National Institutes of Health (R01-EY026031) to Mark A Frye.

## Additional information

### Funding

| Funder | Grant reference number | Author |
|---|---|---|
| National Eye Institute | EY026031 | Mark A Frye |

The funders had no role in study design, data collection and interpretation, or the decision to submit the work for publication.

### Author contributions

Giovanni Frighetto, Conceptualization, Data curation, Software, Formal analysis, Validation, Investigation, Visualization, Methodology, Writing - original draft; Mark A Frye, Conceptualization, Supervision, Funding acquisition, Methodology, Writing - original draft, Project administration, Writing - review and editing

### Author ORCIDs

Giovanni Frighetto http://orcid.org/0000-0001-7461-3916
Mark A Frye http://orcid.org/0000-0003-3277-3094

### Decision letter and Author response

Decision letter https://doi.org/10.7554/eLife.83656.sa1
Author response https://doi.org/10.7554/eLife.83656.sa2

## Additional files

### Supplementary files

• MDAR checklist

### Data availability

Source data plus Matlab and R analysis code for all figures is provided on OSF https://doi.org/10.17605/OSF.IO/C9N4Y.

The following dataset was generated:

| Author(s) | Year | Dataset title | Dataset URL | Database and Identifier |
|---|---|---|---|---|
| Frighetto G, Frye MA | 2023 | Data and Analysis Code from: Columnar neurons support saccadic bar tracking in *Drosophila* | https://doi.org/10.17605/OSF.IO/C9N4Y | Open Science Framework, 10.17605/OSF.IO/C9N4Y |

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
