## [Editor Report]

This study presents valuable new insights into the neural circuits in the fly visual system that mediate object tracking during flight, which have received less prior attention than the circuits involved in motion vision. The evidence supporting the claims of the authors is convincing. The work will be of interest to neuroscientists working on visual neural circuits and visually-guided behavior.

---

## [Decision Letter]

**Decision letter after peer review:**

Thank you for submitting your article "Feature detecting columnar neurons mediate object tracking saccades in *Drosophila*" for consideration by *eLife*. Your article has been reviewed by 3 peer reviewers, and the evaluation has been overseen by a Reviewing Editor and Claude Desplan as the Senior Editor. The reviewers have opted to remain anonymous.

Essential revisions:

1. Additional experimental evidence is needed to resolve whether T3 neurons function as local object detectors. Since all the behavioral experiments in this manuscript use full length bars, it is not possible to know whether the T3 results generalize to small objects. It is also difficult to reconcile the stronger response to small objects with the role ascribed to T3 cells in generating behavioral responses to long bars.

2. Improved quantification of the data in Figure 3 is needed to support the claim that T3-silenced flies don't track the bar anymore, but T4-T5-silenced flies still do despite performing fewer saccades. For example, if the authors included a model for saccade amplitude that draws on both object position and speed (and thus both T3 and T4/T5 pathways) to model the amplitude of the turns, the subtlety of the effects could be explained (and even predicted).

3. It is unclear how prolonged optogenetic activation of all T3 cells would affect the downstream circuitry. It is also unclear that this experiment is equivalent to a "loss-of-function perturbation" of T3 cells, as the authors claim in the text. The conclusions based on these data should be moderated, and the authors' might consider reducing the emphasis on these data within the manuscript.

4. The authors should rectify or justify their selection of genetic control lines for behavioral experiments, specifically the lack of controls that use the parental stocks.

5. The use of statistical tests requires additional clarification.

*Reviewer #1 (Recommendations for the authors):*

1. Generally, I think a better quantification and more in-depth discussion would improve the manuscript (see specific points raised in the general comments and below).

2. Elaborating on point 3 I raised above: If T3 cells responded to flicker, wouldn't you expect tuning to temporal frequency? I find the evidence supporting velocity tuning rather weak. For example in Figure 1_1C the response at 2Hz is larger than for 1Hz at the same speed. Generally, the results are likely confounded by how many T3 cells are active at the same time, which depends on the spatial wavelength of the pattern, if I assume correctly that these data represent population averages and not individual cells. More experiments or a more rigorous analysis is needed to support the claim that T3 cells are tuned to stimulus velocity.

3. Silencing of T4 and T5 cells also leads to a reduction in the number of saccades in the experiments with magnetically tethered flies and to a reduction in saccade amplitude. However, the authors claim that these flies are still able to track the bar. How are the flies able to do that, as they should also not be able to smoothly track bar motion? Is the direction of saccades with relation to bar position different for the different genotypes?

Generally, a better quantification of the fidelity of bar tracking would help to compare the different genotypes. For example, you could quantify the difference between bar position and heading as a measure for tracking fidelity.

*Reviewer #2 (Recommendations for the authors):*

– Line 39

This statement is too strong. Even the cited paper (Busch 2018) states: "Although it is becoming clear that HS cell activity influences locomotor behavior, the underlying biological significance is a different and more difficult question to address."

– Line 89

For imaging the information on the T3 line is sufficient (Keles 2020 shows it is very clean in the optic lobe). However, when the line is used for activation and inactivation behavioral experiments, an image of the entire brain (preferably with the VNC) is necessary.

– Line 113

Mapping procedure is not ideal. Tanaka 2020 map T3 receptive fields with small squares moving vertically and horizontally, but they refrain from multiplying the results since it can lead to an artificially small RF center. In addition, the Tanaka paper shows that long bars suppress responses in T3 cells, which make the mapping stimulus itself less than ideal (although it is clear T3s do respond to bars and the stimulus speed is different in the two papers). Authors should either change the mapping stimulus, add an analysis method that show a level of independence between adjacent ROIs, or change their language to show that they are analyzing ROI which are likely individual neurons.

– Manipulation experiments in Figure 2, 3, 4, and 5 are missing parental controls.

– T4/T5 and T3 crossed to WT or mutated Kir (Kir2.1 nonconducting, e.g. Harrell, J. Neurosci 2021).

– T4/T5 and T3 crossed to WT or no ATR fed flies.

– If the model is included in the paper, it would be interesting to see some predictions from the model that are tested and verified by data. A model fit on its own is of less interest, since it is simply one of many ways of explaining the results.

*Reviewer #3 (Recommendations for the authors):*

Major concern 1.

The three main conclusions the authors are trying to draw from these data are that (1) the T3 feature detecting pathway is responsible for triggering saccades in response to moving objects, but not widefield motion, (2) the T4/T5 pathway is responsible both for triggering saccades in response to widefield motion, and (3) the T4/T5 pathways is responsible for controlling the amplitude of both object and widefield motion triggered saccades (line 269-271: "whereas T4/T5 function is essential to controlling saccade amplitude"). The evidence for the first two points is compelling. The third point, however, is not well supported by the data (or the current presentation of the data).

For example, in Figure 3F, the saccade amplitude is reduced for both the T3sp>kir and T4T5sp>kir. Indicating that both pathways play a role in determining saccade amplitude. What are their relative effect sizes? Second example, in Figure 4E all of the flies (controls, T3sp, T4T5sp) show an increase in saccade amplitude in response to the optogenetic ON stimulus. The effect sizes seem to be quite small. What are the relative effect sizes? i.e., are both of the following two expressions significantly greater than zero:

(a) (Saccade amplitude for T4T5sp>CsC ON – Saccade amplitude for T4T5sp>CsC OFF) -(Saccade amplitude for Esp>CsC ON – Saccade amplitude for Esp>CsC OFF)

(b) (Saccade amplitude for T4T5sp>CsC ON – Saccade amplitude for T4T5sp>CsC OFF) – (Saccade amplitude for T3sp>CsC ON – Saccade amplitude for T3sp >CsC OFF)

Alternatively, if you do a statistical analysis to place all six of the experiments from Figure 4E into groups, i.e. a,b,c, are the T4T5sp>CsC ON flies in their own group?

Are the relative effect sizes meaningful in a behavioral context?

Recommendation: in addition to addressing the statistical questions above, I strongly encourage the authors to put their results into a broader framework. To me it would make sense if the saccade amplitude was driven by both pathways, as there should be both a position and velocity component of the object that drives where the fly turns to. For example, Mongeau and Frye 2017 show that saccade amplitude is related to the pre-saccade error angle. The role of the T4/5 and T3 cells in this behavior is not clear. I believe the authors now have the data to model the individual contributions of T4/5 and T3 to the feedforward motor command at a mechanistic level, perhaps by fitting a proportional-derivative feedforward controller to their data, and compare the results from this model to the results from the silencing experiments. Such an analysis would very nicely compliment the modeling done to explain the saccade triggering.

Major concern 2.

Figure 3B shows that T4T5Sp>Kir flies can still track the motion-defined bar to some extent, but the saccade rate is the same (or even lower) than the T3Sp>Kir flies. When I saw this, I thought that the saccade amplitude and/or duration must have increased to compensate for the lower number of saccades, but these data are actually even lower (Figure 3E-F)! How is this possible? Are T3Sp using more smooth movements, such that they can track the bar, but not with saccades? This would be surprising, because wildtype magnetically tethered flies don't seem to perform much smooth movement at all during bar tracking (Mongeau and Frye 2017). This needs to be clarified.

Major concern 3.

Lines 259-261: I am confused by what is happening in this experiment/analysis: "Yet, when the optogenetic stimulus was switched OFF, recovery from sustained saturating depolarization strongly revived saccadic bar tracking in flies expressing CsChrimson in T3 (Figure 4D)."

I have two questions.

(1) The T3 activated flies don't just show revived saccadic bar tracking, they seem to show potentially overcompensated saccadic bar tracking. Could there be a lasting effect from the Chrimson activation that is leading to some overcompensation? It would be helpful to see this experiment repeated with the same genotype, but where the flies never receive any activation. That would help reveal whether there is a history dependence effect.

(2) Can I compare the control flies' OFF epoch from Figure 4D and the control flies from 3C? They show drastically different saccade rates. Why is this?

---

## [Author Response]

Essential revisions:1. Additional experimental evidence is needed to resolve whether T3 neurons function as local object detectors. Since all the behavioral experiments in this manuscript use full length bars, it is not possible to know whether the T3 results generalize to small objects. It is also difficult to reconcile the stronger response to small objects with the role ascribed to T3 cells in generating behavioral responses to long bars.

In the new version of the manuscript we revised the claim related to T3 as local object detector neurons. We focused our paper on long vertical bars and we addressed the discrepancy with previous work.

2. Improved quantification of the data in Figure 3 is needed to support the claim that T3-silenced flies don't track the bar anymore, but T4-T5-silenced flies still do despite performing fewer saccades. For example, if the authors included a model for saccade amplitude that draws on both object position and speed (and thus both T3 and T4/T5 pathways) to model the amplitude of the turns, the subtlety of the effects could be explained (and even predicted).

We delved further into the bar tracking dynamics of T3>Kir and T4/T5>Kir flies in the magnetically tether setup (Figure 3) to better understand how the latter was still showing good bar tracking performance (the strategy by comparison to T3>Kir is very different). We added new plots to explain how that was achieved and to highlight the different roles of T3 and T4/T5 in bar tracking behavior.

3. It is unclear how prolonged optogenetic activation of all T3 cells would affect the downstream circuitry. It is also unclear that this experiment is equivalent to a "loss-of-function perturbation" of T3 cells, as the authors claim in the text. The conclusions based on these data should be moderated, and the authors' might consider reducing the emphasis on these data within the manuscript.

We added clarification for our interpretation of “loss-of-function perturbation” and, at the same time, we tempered the conclusions from this experiment.

4. The authors should rectify or justify their selection of genetic control lines for behavioral experiments, specifically the lack of controls that use the parental stocks.

We explained our methodological approach for genetic controls, which is an emerging technology.

5. The use of statistical tests requires additional clarification.

We added more information in figure captions and better explained our statistical approach in the methods section.

Reviewer #1 (Recommendations for the authors):1. Generally, I think a better quantification and more in-depth discussion would improve the manuscript (see specific points raised in the general comments and below).

In the new version of the manuscript, we expanded the quantification with more analyses and the discussion with more thorough explanations.

2. Elaborating on point 3 I raised above: If T3 cells responded to flicker, wouldn't you expect tuning to temporal frequency? I find the evidence supporting velocity tuning rather weak. For example in Figure 1_1C the response at 2Hz is larger than for 1Hz at the same speed. Generally, the results are likely confounded by how many T3 cells are active at the same time, which depends on the spatial wavelength of the pattern, if I assume correctly that these data represent population averages and not individual cells. More experiments or a more rigorous analysis is needed to support the claim that T3 cells are tuned to stimulus velocity.

Yes, we would expect temporal frequency tuning but if we add the spatial structure sensitivity that only full-wave rectified neurons can enable (i.e., the response to each change in luminance contrast), the result is a tuning to the temporal frequency biased by the spatial structure of the stimulus which essentially means speed-tuning. Taking into account the three features that we have shown: (a) full-wave rectified; (b) directional agnostic; (c) response to high spatial frequency stimuli (i.e., motion-defined bars); it is enough to consider T3 neurons as the perfect candidates for speed detection.

The responses do not represent population averages but mean responses of a single neuron per fly (at least we tried to draw a single ROI per fly around a single presynaptic terminal of a T3 neuron in the lobula).

In the new version of the manuscript we included a spatiotemporal plot of a broader range of temporal and spatial frequencies in order to highlight the speed-tuning property of T3.

3. Silencing of T4 and T5 cells also leads to a reduction in the number of saccades in the experiments with magnetically tethered flies and to a reduction in saccade amplitude. However, the authors claim that these flies are still able to track the bar. How are the flies able to do that, as they should also not be able to smoothly track bar motion? Is the direction of saccades with relation to bar position different for the different genotypes?Generally, a better quantification of the fidelity of bar tracking would help to compare the different genotypes. For example, you could quantify the difference between bar position and heading as a measure for tracking fidelity.

We added more analyses to show how T4/T5 silenced flies were still able to track the bar without making as many saccades as control flies, rather they smoothly track the bar as they are less able to stabilize their gaze against the large-field ground. We included a performance index to compare the bar tracking fidelity in the three genotypes either in silencing and activation experiments.

Reviewer #2 (Recommendations for the authors):Major points– Line 39This statement is too strong. Even the cited paper (Busch 2018) states: "Although it is becoming clear that HS cell activity influences locomotor behavior, the underlying biological significance is a different and more difficult question to address."

We toned down the statement.

– Line 89For imaging the information on the T3 line is sufficient (Keles 2020 shows it is very clean in the optic lobe). However, when the line is used for activation and inactivation behavioral experiments, an image of the entire brain (preferably with the VNC) is necessary.

We added new confocal images of the entire nervous system (see new Figure 3 —figure supplement 1).

– Line 113Mapping procedure is not ideal. Tanaka 2020 map T3 receptive fields with small squares moving vertically and horizontally, but they refrain from multiplying the results since it can lead to an artificially small RF center. In addition, the Tanaka paper shows that long bars suppress responses in T3 cells, which make the mapping stimulus itself less than ideal (although it is clear T3s do respond to bars and the stimulus speed is different in the two papers). Authors should either change the mapping stimulus, add an analysis method that show a level of independence between adjacent ROIs, or change their language to show that they are analyzing ROI which are likely individual neurons.

We stressed the fact that we analyzed single neurons.

– Manipulation experiments in Figure 2, 3, 4, and 5 are missing parental controls.

We think that empty-splitGal4 crossed with the reporter is a better control line to use than the parental lines (see for some example: Klapoetke et al. 2017; Namiki et al. 2022; Ache et al. 2019). We explained that in the new version of the manuscript.

– T4/T5 and T3 crossed to WT or mutated Kir (Kir2.1 nonconducting, e.g. Harrell, J. Neurosci 2021).– T4/T5 and T3 crossed to WT or no ATR fed flies.– If the model is included in the paper, it would be interesting to see some predictions from the model that are tested and verified by data. A model fit on its own is of less interest, since it is simply one of many ways of explaining the results.

The model we proposed it is not just a fit of the data: (i) we provided a plausible biological substrate (T3 neurons) to a model for interceptive saccade exclusively based on behavioral data; (ii) we put forward a mechanism that predicts the time window needed by the integrator (strictly related to the calcium dynamics); (iii) we predicted the number of cells (in 1D) that would be necessary to trigger a saccade that can be tested in future experiments using holographic optogenetic stimulation of T3 neurons. We added these and other predictions in the new version of the manuscript.

Reviewer #3 (Recommendations for the authors):Major concern 1.The three main conclusions the authors are trying to draw from these data are that (1) the T3 feature detecting pathway is responsible for triggering saccades in response to moving objects, but not widefield motion, (2) the T4/T5 pathway is responsible both for triggering saccades in response to widefield motion, and (3) the T4/T5 pathways is responsible for controlling the amplitude of both object and widefield motion triggered saccades (line 269-271: "whereas T4/T5 function is essential to controlling saccade amplitude"). The evidence for the first two points is compelling. The third point, however, is not well supported by the data (or the current presentation of the data).For example, in Figure 3F, the saccade amplitude is reduced for both the T3sp>kir and T4T5sp>kir. Indicating that both pathways play a role in determining saccade amplitude. What are their relative effect sizes? Second example, in Figure 4E all of the flies (controls, T3sp, T4T5sp) show an increase in saccade amplitude in response to the optogenetic ON stimulus. The effect sizes seem to be quite small. What are the relative effect sizes? i.e., are both of the following two expressions significantly greater than zero:(a) (Saccade amplitude for T4T5sp>CsC ON – Saccade amplitude for T4T5sp>CsC OFF) -(Saccade amplitude for Esp>CsC ON – Saccade amplitude for Esp>CsC OFF)(b) (Saccade amplitude for T4T5sp>CsC ON – Saccade amplitude for T4T5sp>CsC OFF) – (Saccade amplitude for T3sp>CsC ON – Saccade amplitude for T3sp >CsC OFF)

In the new version of the manuscript we fixed some data fitting to accommodate the long tail distributions of saccade amplitude and duration. Then, we computed standardized effect sizes (Cohen’s d) for all our comparisons. Moreover, we decided to reduce the emphasis on the saccade dynamics results since they were not as compelling as the saccade frequency.

Alternatively, if you do a statistical analysis to place all six of the experiments from Figure 4E into groups, i.e. a,b,c, are the T4T5sp>CsC ON flies in their own group?

If Reviewer #3 is asking whether the comparisons were made within groups, the answer is yes.

Are the relative effect sizes meaningful in a behavioral context?

Generally speaking they are. However, as aforementioned we reduced the importance of these specific results.

Recommendation: in addition to addressing the statistical questions above, I strongly encourage the authors to put their results into a broader framework. To me it would make sense if the saccade amplitude was driven by both pathways, as there should be both a position and velocity component of the object that drives where the fly turns to. For example, Mongeau and Frye 2017 show that saccade amplitude is related to the pre-saccade error angle. The role of the T4/5 and T3 cells in this behavior is not clear. I believe the authors now have the data to model the individual contributions of T4/5 and T3 to the feedforward motor command at a mechanistic level, perhaps by fitting a proportional-derivative feedforward controller to their data, and compare the results from this model to the results from the silencing experiments. Such an analysis would very nicely compliment the modeling done to explain the saccade triggering.

In Mongeau and Frye 2017, saccade amplitude is weakly related to bar speed, whereas pre-saccadic error angle (which is related to bar speed) and post-saccadic error angle are strongly correlated, meaning that saccade amplitude is pretty stereotyped and mainly speed independent. On the contrary, inter-saccadic interval is related to bar speed, meaning that if the bar speed increases, flies chase the bar by increasing the saccade frequency but keeping quite constant their amplitude. We put forward a model in which T3 control the triggering mechanism (by integrating the bar displacement over a fixed amount of time) and T4/T5 maintain the gaze stabilization through parallel pathways. In this sense, the model based on T3 excludes any pure position or velocity terms and a PD controller is not probably a good description of the biological control system.

Major concern 2.Figure 3B shows that T4T5Sp>Kir flies can still track the motion-defined bar to some extent, but the saccade rate is the same (or even lower) than the T3Sp>Kir flies. When I saw this, I thought that the saccade amplitude and/or duration must have increased to compensate for the lower number of saccades, but these data are actually even lower (Figure 3E-F)! How is this possible? Are T3Sp using more smooth movements, such that they can track the bar, but not with saccades? This would be surprising, because wildtype magnetically tethered flies don't seem to perform much smooth movement at all during bar tracking (Mongeau and Frye 2017). This needs to be clarified.

We added new data to show that T4/T5Sp>Kir2.1 flies used smooth movement to track the bar (see new Figure 4). Our interpretation is that the number of saccades was reduced because the pre-saccadic error angles were not sufficiently large to trigger saccades. This is likely the result of the flies’ inability to properly stabilize the background. We discussed this in our revision.

Major concern 3.Lines 259-261: I am confused by what is happening in this experiment/analysis: "Yet, when the optogenetic stimulus was switched OFF, recovery from sustained saturating depolarization strongly revived saccadic bar tracking in flies expressing CsChrimson in T3 (Figure 4D)."I have two questions.(1) The T3 activated flies don't just show revived saccadic bar tracking, they seem to show potentially overcompensated saccadic bar tracking. Could there be a lasting effect from the Chrimson activation that is leading to some overcompensation? It would be helpful to see this experiment repeated with the same genotype, but where the flies never receive any activation. That would help reveal whether there is a history dependence effect.(2) Can I compare the control flies' OFF epoch from Figure 4D and the control flies from 3C? They show drastically different saccade rates. Why is this?

(1) We added new analysis and data addressing this concern (new Figure 5 —figure supplement 2 and Figure 6 —figure supplement 3).

(2) No, but you can divide the number of saccade in old Figure 3C by 2 cause in this case the number of saccades are spread across 25 s while in old Figure 4D the OFF period is on average half that time (ON + OFF periods = 25 s). Also, experiments were run in two different setups that might have generated a relative number of saccades due to slightly different arrangement of the magnets. In the new version of the manuscript we use saccade frequency instead of number to have easier comparisons.